# THERE IS NO VAE: END-TO-END PIXEL-SPACE GENERATIVE MODELING VIA SELF-SUPERVISED PRE-TRAINING

**Jiachen Lei**[1]  **Keli Liu**[1]  **Julius Berner**[2]  **Haiming Yu**[1]  **Hongkai Zheng**[2]
**Jiahong Wu**[1*]  **Xiangxiang Chu**[1]
[1] AMAP, Alibaba Group, [2] Caltech

## ABSTRACT

Pixel-space generative models are often more difficult to train and generally underperform compared to their latent-space counterparts, leaving a persistent performance and efficiency gap. In this paper, we introduce a novel two-stage training framework that closes this gap for pixel-space diffusion and consistency models. In the first stage, we pre-train encoders to capture meaningful semantics from clean images while aligning them with points along the same deterministic sampling trajectory, which evolves points from the prior to the data distribution. In the second stage, we integrate the encoder with a randomly initialized decoder and fine-tune the complete model end-to-end for both diffusion and consistency models. Our framework achieves state-of-the-art (SOTA) performance on ImageNet. Specifically, our diffusion model reaches an FID of 1.58 on ImageNet-256 and 2.35 on ImageNet-512 with 75 number of function evaluations (NFE) surpassing prior pixel-space methods and VAE-based counterparts by a large margin in both generation quality and training efficiency. In a direct comparison, our model significantly outperforms DiT while using only around 30% of its training compute. Furthermore, our consistency model achieves an impressive FID of 8.82 on ImageNet-256, significantly outperforming its latent-space counterparts. This marks the first successful training of a consistency model directly on high-resolution images without relying on pre-trained VAEs or diffusion models. Our codes are available at: https://github.com/AMAP-ML/EPG

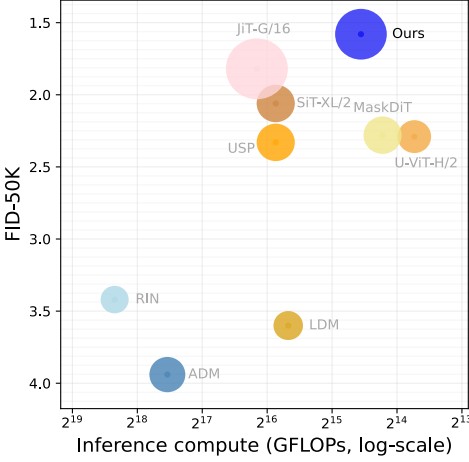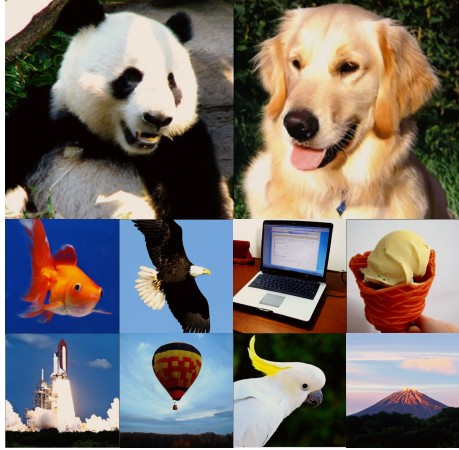

Figure 1: **(Left)** Our diffusion model achieves SOTA performance while maintaining significant inference efficiency. The $x$-axis indicates the log-scaled GFLOPs for generating an image. The bubble size denotes number of model parameters. **(Right)** Images generated by our diffusion model.

---

*Corresponding Author

# 1    INTRODUCTION

Diffusion-based generative models have become the cornerstone of modern image synthesis. This family includes both traditional, iterative diffusion models (Ho et al., 2020; Song et al., 2021; Karras et al., 2022) and more recent one/few-shot generators like consistency models (Song et al., 2023; Geng et al., 2025; Lu & Song, 2025) and flow maps (Boffi et al., 2025). Much of their success on high-resolution tasks relies on training in a compressed latent space (Rombach et al., 2022) of pre-trained VAEs (Kingma & Welling, 2013). However, this reliance on VAEs introduces its own set of significant challenges. Firstly, training the VAE itself is difficult due to the need to balance compression with high-fidelity reconstruction. Besides, even when trained properly, the VAE often produces imperfect reconstructions for latents far from the training set. While pre-training the VAE on massive datasets can mitigate this, it still induces a permanent performance bottleneck, as the generative model's ability to adapt to new data is always limited by the VAE's fixed capacity.

To simplify the overall training pipeline and bypass the performance bottleneck of VAEs, numerous works (Jabri et al., 2023; Hoogeboom et al., 2023; Ho et al., 2021; Dhariwal & Nichol, 2021) have explored training diffusion models directly on raw pixels, developing specialized model architectures (Jabri et al., 2023) or training techniques (Hoogeboom et al., 2023). Despite these efforts, none have achieved training performance and inference efficiency comparable to VAE-based methods primarily due to the high computational cost originating from the model backbone or the slow convergence rate, representing the two major challenges of operating in pixel space.

In our work, we aim at addressing these challenges to bridge the performance and efficiency gap between pixel-space training and its latent-space counterparts. We take inspiration from self-supervised learning (SSL) approaches: the encoder served as general visual semantic learner and decoder as task-specific prediction head (He et al., 2021; Chen et al., 2021; Chen & He, 2020). This decomposition significantly improves training efficiency and model performance in downstream tasks. Motivated by this, we hypothesize that the encoder-decoder role in diffusion-based generative models can also be decomposed in a similar way. Specifically, we argue encoders in diffusion-based generative models primarily learn high-level visual semantics from noisy inputs and the decoders act as low-level pixel generators conditioned on encoder representations.

As a result, we decompose the training paradigm into two separate stages as in SSL. During pre-training, the encoder captures meaningful visual semantics from images at varying noise levels. Note that this is different from the Gaussian noise data augmentation utilized in SSL, as the noise level follows pre-defined diffusion schedules and is much higher than the one used in augmentation. Subsequently, after pre-training, the decoder is fine-tuned end-to-end with the encoder under task-specific configurations and predicts pixels given representations from the encoder. This framework endows the model with solid discriminative capability at the start of training, while offering a more flexible way for it to adapt learned representations to contain detailed visual semantics, which is preferable in generative tasks. However, designing such a pre-training method is non-trivial, as typical visual learning methods struggle in learning meaningful semantics from images with larger noise levels. In particular, directly utilizing SSL as a pre-training method is ineffective, due to its representation collapse on images of strong noise.

To address this, by extending the idea of rRCM (Lei et al., 2025), we pre-train encoders to capture visual semantics from clean images while aligning them with corresponding points on the same deterministic sampling trajectory, which evolves points from pure Gaussian to the data distribution. In practice, this is realized by matching images with shared noise but different noise levels as in consistency tuning (Song et al., 2023). This pre-training approach reformulates representation learning on noisy images as a generative alignment task, connecting features of noisy samples to their progressively cleaner versions. Subsequently, given the pre-trained encoders, we fine-tune it alongside a randomly initialized decoder in an end-to-end manner under task-specific configurations.

To verify the effectiveness of our method, we conduct thorough experiments on ImageNet dataset without relying on any external models. We focus on two distinct downstream generative tasks: training diffusion and consistency models. Built upon pre-trained weights, our diffusion model achieves a SOTA FID of 1.58 on ImageNet-256 and 2.35 on ImageNet-512 with 75 NFEs, surpassing prior and concurrent pixel-space methods by a large margin in both generation quality and efficiency. Besides, we close the gap with the dominant latent-space paradigm and achieve performance that surpasses leading latent diffusion counterparts, such as DiT (Peebles & Xie, 2023) and

SiT (Ma et al., 2024), while using approximately 30% of its training compute. Moreover, our consistency model achieves a remarkable FID score of 8.82 in a single generation step, demonstrating superior performance and training efficiency compared to the latent-space counterparts. This result, to the best of our knowledge, marks the first time a consistency model has been successfully trained directly on ImageNet-256 without utilizing pre-trained VAEs or diffusion models. While we rely on an SSL pre-training, this is significantly more efficient and shows a promising way of stabilizing consistency model training.

Our contributions can be summarized as follows:

i. We propose a novel training framework that enables efficient, performant, and scalable pixel-space generative modeling in high resolution. By identifying the semantic roles of the encoder and decoder, we establish that training a diffusion model can be framed as a self-supervised learning problem, similar to training an image classifier. Moreover, our empirical results validate that the key to successful pixel-space generation is the high-quality semantic representation that is temporally consistent across noise levels.

ii. Our *End-to-end Pixel-space Generative model* (**EPG**) demonstrates state-of-the-art results for pixel-space generation on ImageNet dataset. For the first time, it closes the performance and efficiency gap with leading VAE-based counterparts. Besides, to the best of our knowledge, we are the first to achieve strong generation performance on ImageNet-256 with consistency models directly trained in pixel space and without relying on pre-trained VAEs or diffusion models.

iii. In terms of both GFLOPs and training time, our model, built on Vision Transformer (Vaswani et al., 2017), maintains significant computational efficiency across different resolutions. This is achieved by fixing input token length through proportionally adjusting the patch size as image resolution increases e.g., 16×16 on ImageNet-256 and 32×32 on ImageNet-512.

## 2 BACKGROUND

In this section, we introduce necessary backgrounds on diffusion models and consistency models. Let $p_{data}(\boldsymbol{x})$ denote the probability density of target data $\boldsymbol{x}$, $p_t(\boldsymbol{x})$ the probability density of $\boldsymbol{x}(t)$, $\boldsymbol{x}(0) \sim p_{data}(\boldsymbol{x})$. Diffusion models (DM) approximate $p_{data}$ by learning to reverse a pre-defined forward stochastic differential equation (SDE) that gradually transports points $\boldsymbol{x}(0)$ from $p_{data}(\boldsymbol{x})$ to a prior distribution. The corresponding reverse SDE generate samples by evolving points from the prior distribution back to the data distribution, guided by the score function $\nabla_{\boldsymbol{x}} \log p_t(\boldsymbol{x})$. To estimate the score, diffusion models train a neural network $s_\theta(\boldsymbol{x}(t), t)$ using the objective (Karras et al., 2022)

$$\arg\min_\theta \mathbb{E}\left[\lambda(t) \|s_\theta(\boldsymbol{x}(t), t) - \boldsymbol{x}(0)\|^2\right]. \tag{1}$$

Here, $\lambda(t)$ is a scalar weighting function. In our work, following EDM (Karras et al., 2022), we define the forward process as $d\boldsymbol{x} = \sqrt{2t}d\boldsymbol{w}$, $t \in [0, T], T = 80$. $\boldsymbol{w}$ is the standard Wiener process. The corresponding reverse SDE can be depicted by: $d\boldsymbol{x} = -2t\nabla_{\boldsymbol{x}} \log p_t(\boldsymbol{x}) + \sqrt{2t}d\bar{\boldsymbol{w}}$ with $\bar{\boldsymbol{w}}$ being a standard Wiener process that flows backwards in time. A key property of the reverse SDE is the existence of the probability flow (PF) ordinary differential equation (ODE) that shares the same marginal distribution $p_t(\boldsymbol{x})$

$$d\boldsymbol{x} = -t\nabla_{\boldsymbol{x}} \log p_t(\boldsymbol{x}). \tag{2}$$

Let $f(\boldsymbol{x}(t), t)$ denote the solution of equation 2 from $t$ to 0

$$f(\boldsymbol{x}(t), t) = \boldsymbol{x}(t) + \int_t^0 -s\nabla_{\boldsymbol{x}} \log p_s(\boldsymbol{x})ds. \tag{3}$$

Diffusion models generate samples by solving equation 3 numerically over discrete time intervals. As a result, it requires multiple function evaluations, incurring heavy computational cost.

Consistency models (CM) (Song et al., 2023) address this sampling inefficiency through directly approximating the trajectory end point $f(\boldsymbol{x}(t), t)$ with a neural network $f_\theta(\boldsymbol{x}(t), t)$, which is trained by enforcing the self-consistency condition: $f_\theta(\boldsymbol{x}(t), t) = f_\theta(\boldsymbol{x}(s), s), t, s \in [0, T]$. Note that $\boldsymbol{x}(t)$

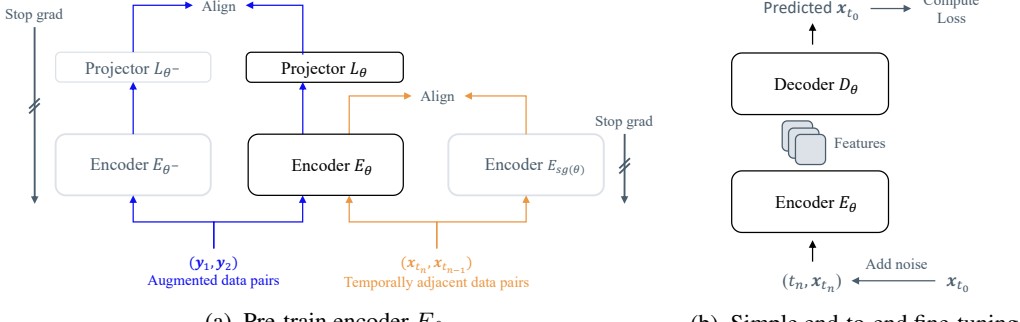

(a) Pre-train encoder $E_\theta$.     (b) Simple end-to-end fine-tuning.

Figure 2: Overview of our framework. **During pre-training**, we train encoder by learning visual semantics (blue branch) and aligning them along ODE sampling trajectories (yellow branch). $\theta^-$ is the exponential moving average of $\theta$, and $sg$ is the stop gradient operation. **After pre-training**, we discard the projector and train $E_\theta$ along side a randomly initialized decoder $D_\theta$ in an end-to-end manner under specific generative training settings (e.g., denoising or consistency training). The encoder $E_\theta$ maps noisy images into features, which are then reconstructed into clean pixels by the decoder $D_\theta$. In our work, we choose vision transformer as our backbone.

and $\boldsymbol{x}(s)$ lie on the same PF ODE trajectory and $f_\theta$ satisfies the boundary condition: $f_\theta(\boldsymbol{x}(0), 0) = \boldsymbol{x}(0)$. To ensure this, prior works parameterize $f_\theta$ as

$$f_\theta(\boldsymbol{x}(t), t) = c_{skip}(t)\boldsymbol{x}(t) + c_{out}(t)F_\theta(\boldsymbol{x}(t), t), \tag{4}$$

where $c_{skip}(t)$ and $c_{out}(t)$ are scalar functions and satisfy $c_{skip}(0) = 1, c_{out}(0) = 0$. CMs discretize the time horizon into $N-1$ non-overlapping time intervals $\{t_n\}_{n=0}^{N-1}, t_n \in [\sigma_{min}{}^*, T]$, and optimize the following *consistency training* objective, which minimizes a metric between adjacent points on the PF ODE sampling trajectory

$$\arg\min_\theta \mathbb{E}\big[\lambda(t)\boldsymbol{d}(f_\theta(\boldsymbol{x}_{t_n}, t_n), f_{\boldsymbol{sg}(\theta)}(\boldsymbol{x}_{t_{n-1}}, t_{n-1}))\big], \tag{5}$$

where $\boldsymbol{d}$ is a distance metric, $\boldsymbol{sg}$ means stop gradient operation, $\boldsymbol{x}_{t_n}, \boldsymbol{x}_{t_{n-1}}$ are adjacent points on the same PF ODE trajectory. During training, $\boldsymbol{x}_{t_n}$ is sampled from the forward SDE given clean image $\boldsymbol{x}_{t_0}$. $\boldsymbol{x}_{t_{n-1}}$ is sampled by following equation 2, yet the score at $\boldsymbol{x}_{t_n}$ is necessary but unknown during training. To address this, one can either utilize a pre-trained diffusion model, or leverage the unbiased estimator $-\mathbb{E}[\frac{\boldsymbol{x}_t - \boldsymbol{x}}{t^2}|\boldsymbol{x}_t]$ (Song et al., 2023) to approximate the score value, meaning that $\nabla_{\boldsymbol{x}} \log p_t(\boldsymbol{x}) \approx -(\boldsymbol{x}_t - \boldsymbol{x})/t^2$. As a result, according to equation 2, $\boldsymbol{x}_{t_{n-1}}$ can be approximated by

$$\boldsymbol{x}_{t_{n-1}} = \boldsymbol{x}_{t_0} + t_{n-1}\boldsymbol{\epsilon}, \tag{6}$$

where the noise $\boldsymbol{\epsilon}$ is the same perturbation used when creating $\boldsymbol{x}_{t_n}$. In our work, we sample $\boldsymbol{x}_{t_{n-1}}$ by following equation 6 without relying on any trained diffusion models. To further improve model performance, iCT (Song & Dhariwal, 2023) proposes annealing the discretization steps $N$ following a pre-defined discretization schedule, which increases $N$ step-wisely according to training steps. With an initially small $N$, this approach reduces the gradient variance at higher noise levels and accelerate the model convergence speed at small diffusion time steps, providing solid supervising signals for aligning predictions on images of strong noise magnitude.

ECT (Geng et al., 2025) extends the above framework by additionally taking advantage of trained diffusion model weights to initialize $f_\theta$, and trains $f_\theta$ on continuous time intervals. To produce paired noisy samples $(\boldsymbol{x}(t), \boldsymbol{x}(r))$, $r, t \in [\sigma_{min}, T]$, they sample $t$ from a predefined distribution $p(t)$ and derive $r$ with a deterministic mapping function $p(r|t)$. As training proceeds, the ratio $r/t$ gradually increases and approximates 1.0. This reduces early-stage variance of equation 5, while reducing the bias as $r/t$ becomes larger, ultimately improving model generation quality.

---

*The $\sigma_{min}$ is chosen such that $p_{\sigma_{min}}(\boldsymbol{x}) \approx p_{data}(\boldsymbol{x})$ and $\boldsymbol{x}_{t_0}$ can be approximately seen as samples from the data distribution $p_{data}(\boldsymbol{x})$.

## 3 METHOD

### 3.1 OVERVIEW

In this section, we first formalize the core principles of our pre-training method and subsequently introduce the fine-tuning stage. Building upon the diffusion framework established in Section 2, we adhere to the SDE formulations and notational conventions defined therein.

### 3.2 REPRESENTATION CONSISTENCY LEARNING

Specifically, the pre-training objective is composed of two components: a *contrastive loss* that facilitates semantic learning, and a *representation consistency loss* that enforces alignment of semantics across points on the same ODE trajectory. Both objectives are in the form of the equation 5, while adopting the NT-Xent (Chen et al., 2020) as the distance metric. Formally, the NT-Xent metric for both terms can be written as

$$d_{\text{NT-Xent}}(\boldsymbol{q}, \boldsymbol{q}^+) = -\log \frac{\exp\left(\boldsymbol{q} \cdot \boldsymbol{q}^+ / \tau\right)}{\exp\left(\boldsymbol{q} \cdot \boldsymbol{q}^+ / \tau\right) + \sum_{\boldsymbol{q}^-} \exp\left(\boldsymbol{q} \cdot \boldsymbol{q}^- / \tau\right)}, \tag{7}$$

where $\boldsymbol{q}$ is an encoded sample, with $\boldsymbol{q}^+$ and $\boldsymbol{q}^-$ representing its positive and negative counterparts respectively. For the contrastive loss, the positive pairs are constructed through data augmentation, while other samples in the batch are treated as negatives. The representation consistency loss uses temporally adjacent points along the ODE trajectory as positive pairs, e.g., $(\boldsymbol{x}_{t_n}, \boldsymbol{x}_{t_{n-1}})$, while adjacent points from different trajectories serve as negatives, e.g., $(\boldsymbol{x}_{t_n}, \boldsymbol{x}'_{t_{n-1}})$. The temporal data pairs is crafted following equation 6 as discussed in Section 2. As a result, the overall pre-training objective can be formally depicted as

$$\mathbb{E}\left[\underbrace{\boldsymbol{d}_{\text{NT-Xent}}(L_\theta(E_\theta(\boldsymbol{y}_1, t_0)), L_{\theta^-}(E_{\theta^-}(\boldsymbol{y}_2, t_0)))}_{\text{Contrastive Loss}} + \underbrace{\boldsymbol{d}_{\text{NT-Xent}}(E_\theta(\boldsymbol{x}_{t_n}, t_n), E_{\boldsymbol{sg}(\theta)}(\boldsymbol{x}_{t_{n-1}}, t_{n-1}))}_{\text{Representation Consistency Loss}}\right], \tag{8}$$

where $\theta$ denotes model parameter and is updated to minimize the above objective, $(\boldsymbol{y}_1, \boldsymbol{y}_2)$ are augmented views of $\boldsymbol{x}$, $E_{\theta^-}$ and $E_{\boldsymbol{sg}(\theta)}$ are target models used when computing corresponding loss term. $\theta^-$ denotes exponential moving average of $\theta$, respectively. $L_\theta$ is projector layer, a 3-layer MLP, $n \sim \mathcal{U}(0, N)$, and $\lambda(t) = 1$.

Figure 2a illustrates the pre-training framework, which features three branches: a semantic learning branch ($E_{\theta^-}$ and $L_{\theta^-}$), an online branch ($E_\theta$ and $L_\theta$), and a semantic alignment branch ($E_{sg(\theta)}$). The input of each encoder includes a learnable token [CLS], a time condition token, and image tokens. Notably, the online model $E_\theta$ encodes both augmented samples and noisy samples $\boldsymbol{x}_{t_n}$ that contain more noises during each step. When computing losses, both contrastive and representation consistency loss utilize model outputs at the class token position, while contrasting the projector outputs and encoder outputs respectively.

A core design of the original pre-training framework involves annealing EMA coefficient of $\theta^-$ following a manually designed schedule. This mechanism aims to regulate the learning rate of representations on clean images, mitigating the challenges of aligning them with noisy samples at large diffusion time steps, critical for representation quality at high noise levels. This philosophy parallels consistency model training, where consistency at small time steps governs behavior at larger steps (Song & Dhariwal, 2023). However, this approach introduces intricately coupled hyperparameters and a brittle training process, where even minor deviations from the prescribed configuration risk training collapse. As a result, such fragility poses a significant barrier to adaptation in image generation tasks, where achieving better generation quality demands greater hyperparameter flexibility.

In contrast, the temperature value $\tau$ in representation consistency loss intuitively plays a pivotal role in determining the learned representation quality on noisy images. Specifically, a small $\tau$ enforces strong separability between samples on distinct PF ODE trajectories while tightly aligning points within the same trajectory to their clean-image endpoints. To begin with, we set the temperature value to $0.1$, achieving downstream performance comparable to our final results However, its early training exhibits transient instability. We speculate this is because the gradient signals from noisy

samples introduces bias as the model lacks meaningful features to reconcile alignment. To mitigate this, we implement a linear interpolated temperature schedule: $\tau(t) = \tau_1 * (1 - t) + \tau_2 * t$, with $\tau_1 \leq \tau_2, t \in [0, 1]$, leading to a loose alignment for points at large time steps. As training progresses, $\tau_2$ converges to $\tau_1$ via a cosine schedule. Crucially, this schedule operates independently of other hyper-parameters. While this design resolves potential early-stage instability, we emphasize that a fixed small temperature value (e.g., 0.1) remains a viable solution and acts as strong baseline. We ablate this design in Appendix C. We also study the learned semantics of pre-trained model on noisy images (See Appendix E).

### 3.3 Fine-tuning

After pre-training, the projector is discarded and we combine $E_\theta$ with a randomly initialized decoder $D_\theta$ and then fine-tune the complete model $f_\theta$ end-to-end under either diffusion or consistency model training configurations (see Figure 2b).

**Diffusion model.** We fine-tune the complete model with the denoising objective in equation 1. We also incorporate time-dependent weighting function and sample noise levels following a LogNormal distribution. We choose diffusion model, instead of flow matching (Liu et al., 2022; Lipman et al., 2023), as it works seamlessly with our pre-training method, which builds on the theoretical foundations of consistency models (Song et al., 2023). We argue that our pre-training model also supports flow matching in downstream task as the time condition and noise schedules are properly aligned, akin to the recent effort (Lu & Song, 2025).

**Consistency model.** In early experiments, we observe standard consistency training suffers from slow convergence and suboptimal generation quality due to the supervising signals only come from the clean data. To address this, we empirically introduce an auxiliary loss between the model outputs $f_\theta(\boldsymbol{x}_{t_n}, t_n)$ and their corresponding clean images $\boldsymbol{x}_0$ used to generate the noisy inputs $x_{t_n}$. It can be formally depicted by

$$\arg\min_\theta \mathbb{E}\left[\boldsymbol{d}_{\text{NT-Xent}}(W_\phi(f_\theta(\boldsymbol{x}_{t_n}, t_n), t_n), W_\phi(\boldsymbol{x}_{t_0}, t_0))\right]. \tag{9}$$

Here, $W_\phi$ is a frozen copy of the pre-trained encoder $E_\theta$ (without projector layer $L_\theta$) and won't be updated during fine-tuning. This approach provides complementary supervision when training consistency model while incurring negligible cost, and has a similar spirit to the concurrent works (Stoica et al., 2025; Wang & He, 2025). Note that we do not use any external models. Instead, we take full advantage of our pre-trained weights. As a result, we train consistency models by combining equation 5 and equation 9.

We defer detailed diffusion and consistency model training settings (e.g. weighting, time proposal distribution, parameterization, etc.) to Appendix B.2.

## 4 Experiment

**Dataset and network architecture.** We conduct experiments on ImageNet-1K (Deng et al., 2009) dataset. We use vision transformer (ViT) (Vaswani et al., 2017) as our backbone, and utilize $16 \times 16$ patch size on ImageNet-256 and $32 \times 32$ on ImageNet-512. The model input tokens include a learnable token [CLS], time token, and image tokens. During fine-tuning, we use same number of blocks in encoder and decoder. To further improve training efficiency in pixel space, we add residual connections between encoder and decoder as in (Hoogeboom et al., 2023; Bao et al., 2023), and additionally incorporate time condition into decoder using adaLN-Zero (Peebles & Xie, 2023). We defer our network configuration to the Appendix B.3.

**Training.** In main results (Table 1 2 4), we use 1024 batch size in both pre-training and fine-tuning. We pre-train our model for 600K steps (480 epochs). Besides, during fine-tuning, we train diffusion models for 1M steps (800 epochs), and train consistency models for 700K steps (560 epochs). We train models in FP16 mixed precision mode, and use similar hyper-parameter settings across experiments within each training stage. More implementation details, including hyper-parameters and computational cost, are deferred to Appendix B.2. We also display more qualitative results in Appendix F.

Table 1: System-level comparison on ImageNet-256 with CFG. For latent-space models, we display model parameters and sampling GFLOPs of both the VAE and the generative model. We report GFLOPs of our EPG following DiT. [†]: both model parameters and GFLOPs are composed of two generative models and one classifier. Text in gray: method that requires external models in addition to VAE. Line in blue : adopt class-balanced sampling.

| Model | FID↓ | IS↑ | Precision↑ | Recall↑ | NFE↓ | Epochs | #Params | GFLOPs↓ |
|---|---|---|---|---|---|---|---|---|
| ***Models in Latent Space*** | | | | | | | | |
| LDM (Rombach et al., 2022) | 3.60 | 247.7 | **0.87** | 0.48 | 250×2 | 167 | 55 + 400M | 336 + 104 |
| USP DiT-XL/2 (Chu et al., 2025) | 2.33 | 267.0 | - | - | 250×2 | 240 | 84 + 675M | 312 + 119 |
| MaskDiT (Zheng et al., 2024) | 2.28 | 276.6 | 0.80 | 0.61 | 79×2 | 1600 | 84 + 675M | 312 + 119 |
| DiT-XL/2 (Peebles & Xie, 2023) | 2.27 | 278.2 | 0.83 | 0.57 | 250×2 | 1400 | 84 + 675M | 312 + 119 |
| SiT-XL/2 (Ma et al., 2024) | 2.06 | 277.5 | 0.83 | 0.59 | 250×2 | 1400 | 84 + 675M | 312 + 119 |
| REPA (Yu et al., 2025) | 1.42 | 305.7 | 0.80 | 0.65 | 434 | 800 | 84 + 675M | 312 + 119 |
| RAE (Zheng et al., 2025) | 1.28 | 262.9 | - | - | 50×2 | 800 | 415 + 839M | 107 + 146 |
| ***Models in Pixel Space*** | | | | | | | | |
| ADM[†] (Dhariwal & Nichol, 2021) | 3.94 | 215.8 | 0.83 | 0.53 | 500 | 963 | 673M | 761 |
| RIN (Jabri et al., 2023) | 3.42 | 182.0 | - | - | 1000 | 480 | 410M | 334 |
| SiD (Hoogeboom et al., 2023) | 2.44 | 256.3 | - | - | 250×2 | 800 | 2.46B | 555 |
| VDM++ (Kingma & Gao, 2023) | 2.12 | 278.1 | - | - | 250×2 | - | 2.46B | 555 |
| PixNerd-XL/16 (Wang et al., 2025) | 1.93 | 297.0 | 0.79 | 0.59 | 100×2 | 320 | 700M | 134 |
| PixelFlow (Chen et al., 2025c) | 1.98 | 282.1 | 0.81 | 0.60 | - | - | 677M | 2909 |
| SiD2 (Hoogeboom et al., 2025) | 1.72 | - | - | - | - | - | - | 137 |
| **EPG-XL/16** | 2.04 | 283.2 | 0.80 | 0.61 | 75 | 800 | 583M | 128 |
| **EPG-XXL/16** | 1.87 | 287.6 | 0.80 | 0.63 | 75 | 800 | 789M | 176 |
| **EPG-G/16** | 1.70 | 297.8 | 0.80 | 0.63 | 75 | 1600 | 1391M | 321 |
| JiT-H/16 (Li & He, 2025) | 1.86 | **303.4** | 0.78 | 0.62 | 191 | 600 | 953M | 182 |
| JiT-G/16 (Li & He, 2025) | 1.82 | 292.6 | 0.79 | 0.62 | 191 | 600 | 2B | 383 |
| **EPG-XXL/16** | 1.81 | 294.6 | 0.80 | 0.61 | 75 | 600 | 789M | 176 |
| **EPG-G/16** | 1.75 | 275.1 | 0.80 | 0.62 | 75 | 600 | 1391M | 321 |
| **EPG-G/16** | **1.58** | 298.4 | 0.80 | 0.63 | 75 | 1600 | 1391M | 321 |

**Evaluation.** For a fair comparison with existing works, we use evaluation suite provided by ADM (Dhariwal & Nichol, 2021). Evaluation metrics include Frechét Image Distance (FID) (Heusel et al., 2017), sFID (Nash et al., 2021), Inception Score (IS) (Salimans et al., 2016), Precision and Recall (Kynkäänniemi et al., 2019). We report generation results with 50K images sampled by utilizing the Heun ODE sampler (Karras et al., 2022) with 32 sampling steps in Table 3 and additionally with classifier-free guidance (CFG) (Ho & Salimans, 2022) in Table 1 and 2, where we use interval-cfg (Kynkäänniemi et al., 2024) with the recommended guidance interval (0.19, 1.61]. In ablation studies, we sample diffusion models using Euler deterministic sampler with 50 sampling steps, and report FID of diffusion models with 10K images. For consistency models, we report FID with 50K images produced via one-step sampling across all experiments.

**Pre-training cost comparison with VAE**. On ImageNet-256, utilizing 8xH200, we compare our pre-training cost with the sd-vae-mse[†], which is widely adopted in VAE-based methods. It is originally trained for 840K steps in total with a batch size of 192 on large-scale open-sourced datasets. To achieve our performance reported in Table 1, 2, and 4, the pre-training takes 57 hours, 111 hours, and 100 hours respectively. In contrast, it takes 160 hours to train sd-vae-mse. Since our pre-training cost is smaller than the VAE model, we only report our fine-tuning cost (in epochs) when comparing with latent-space methods. We benchmark overall training efficiency against DiT on ImageNet-256 in Table 5.

---

[†]https://huggingface.co/stabilityai/sd-vae-ft-mse

Table 2: System-level comparison on ImageNet-512 with CFG. Settings are the same as Table 1.

| Model | FID↓ | IS↑ | Precision↑ | Recall↑ | NFE↓ | Epochs | #Params | GFLOPs↓ |
|---|---|---|---|---|---|---|---|---|
| *Models in Latent Space* | | | | | | | | |
| U-ViT-H/4 (Bao et al., 2023) | 4.05 | 263.8 | 0.84 | 0.48 | 50×2 | 400 | 84M + 501M | 1260 + 133.0 |
| DiT-XL/2 (Peebles & Xie, 2023) | 3.04 | 240.8 | 0.84 | 0.54 | 250×2 | 600 | 84M + 675M | 1260 + 524.6 |
| SiT-XL/2 (Ma et al., 2024) | 2.62 | 252.2 | 0.84 | 0.57 | 250×2 | 600 | 84M + 675M | 1260 + 524.6 |
| MaskDiT (Zheng et al., 2024) | 2.50 | 256.7 | 0.83 | 0.56 | 79×2 | 800 | 84M + 675M | 1260 + 524.6 |
| *Models in Pixel Space* | | | | | | | | |
| RIN (Jabri et al., 2023) | 3.95 | 216.0 | - | - | 1000 | 800 | 320M | 415 |
| ADM (Dhariwal & Nichol, 2021) | 3.85 | 221.7 | 0.84 | 0.53 | 500 | 1081 | 731M | 2813 |
| SiD (Hoogeboom et al., 2023) | 3.02 | 248.7 | - | - | 250×2 | 800 | 2.46B | - |
| PixelNerd (Wang et al., 2025) | 2.84 | 245.6 | 0.80 | **0.59** | 100×2 | 320 | 700M | 583 |
| VDM++ (Kingma & Gao, 2023) | 2.65 | 278.1 | - | - | 250×2 | 800 | 2.46B | - |
| **EPG-L/32** | 2.43 | 289.7 | 0.82 | 0.57 | 75 | 600 | 540M | 113 |
| **EPG-L/32** | **2.35** | **295.4** | 0.82 | 0.57 | 75 | 800 | 540M | **113** |

Table 3: Class-conditional generation performance comparing with DiT and SiT.

| Model | FID↓ | sFID↓ | IS↑ | Epochs | #Params |
|---|---|---|---|---|---|
| DiT-B/2 | 43.5 | - | - | 80 | 84M+130M |
| SiT-B/2 | 33.0 | 6.46 | 43.71 | 80 | 84M+130M |
| **EPG-B/16** | 52.0 | 8.25 | 24.34 | 80 | 229M |
| **EPG-B/16** | 31.9 | 6.80 | 42.70 | 160 | 229M |
| **EPG-B/16** | **25.1** | **6.27** | **54.25** | 240 | 229M |
| DiT-XL/2 | 19.5 | - | - | 80 | 84M+675M |
| SiT-XL/2 | 17.2 | - | - | 80 | 84M+675M |
| **EPG-XL/16** | **15.9** | 6.28 | 72.43 | 80 | 583M |
| DiT-XL/2 | 9.6 | 6.85 | 121.50 | 1400 | 84M+675M |
| SiT-XL/2 | 8.3 | 6.32 | 131.70 | 1400 | 84M+675M |
| **EPG-XL/16** | **6.6** | **5.16** | **141.09** | 800 | 583M |

Table 4: Benchmarking few-step class-conditional image generation on ImageNet-256.

| Model | FID↓ | NFE↓ | Epochs | #Params |
|---|---|---|---|---|
| *Models in Latent Space* | | | | |
| iCT-XL/2 (Song et al., 2023) | 34.24 | 1 | - | 84M + 675M |
| | 20.30 | 2 | - | 84M + 675M |
| Shortcut-XL/2 (Frans et al., 2025) | 10.60 | 1 | 250 | 84M + 675M |
| | **7.80** | 4 | 250 | 84M + 675M |
| IMM (Zhou et al., 2025) | 8.05 | 1 | 6395 | 84M + 675M |
| *Models in Pixel Space* | | | | |
| **EPG-L/16** | 8.82 | 1 | 560 | 540M |

## 4.1 MAIN RESULTS

**Diffusion Model.** In Table 1, when compared to diffusion models trained in the latent space, EPG-XXL/16 surpasses one of the leading models SiT-XL/2 with similar overall model parameters, e.g., 789M v.s. 759M, and approximately 50% of its training cost. We display the FID convergence of EPG-XL/16 in Figure 7. Our EPG models are in parallel to methods like REPA and RAE and can also benefit from incorporating external supervision, which we leave to our future work. When comparing with pixel-space diffusion models, EPG achieves the best FID while bridging the performance and efficiency gap with VAE-based counterparts. Our EPG models also demonstrate strong scalability with respect to model sizes. By further increasing model parameters to 1391M, our EPG-G/16 model achieves 1.58 FID with class-balanced sampling, outperforming JiT-G/16 by a large margin. We attribute this to the strong semantic quality and consistency from the pre-trained weights. Our performance advantage over JiT also reveals that training diffusion model with the original pixel-wise prediction loss alone is inefficient in capturing rich semantics thus leading to severe overfitting (Li & He, 2025) and poor scalability. We compare class-conditional image generation performance with SiT and DiT of different sizes in Table 3.

EPG's performance on ImageNet-512 (Table 2) confirms its ability to scale efficiently to high-resolution generation. With large 32×32 patch size, it delivers strong generation quality with significantly low computational overhead (GFLOPs), all without relying on external VAEs. The results on ImageNet position EPG as a scalable and efficient solution for pixel-space generation.

**Consistency Model.** To the best of our knowledge, we are the first to successfully train consistency model without relying on pre-trained VAEs or diffusion models. As shown in Table 4, EPG-L/16 reaches 8.82 FID with one step generation, substantially outperforming iCT-XL/2, its latent-space

Table 5: Training efficiency comparison with DiT with 8×H200 GPU. †: measured with official codes.

| Model | FID | Cost (hours) | #Params |
|---|---|---|---|
| sd-vae-mse† | - | 160 | 84M |
| **Our Pre-train** | - | 57 | 106M |
| DiT-XL/2† | 2.27 | 506 | 675M |
| **EPG-XL/16** | 2.04 | 139 | 583M |
| **EPG-XXL/16** | 1.87 | 160 | 789M |

Table 6: Comparing FID scores with baseline methods in downstream tasks.

| Experiment | DM | CM |
|---|---|---|
| REPA SiT-B | 72.71 | - |
| MoCov3 ViT-B | 56.26 | 36.77 |
| Scratch | 59.69 | NaN |
| rRCM | 46.51 | 37.55 |
| **EPG-B/16** | **41.36** | **33.12** |

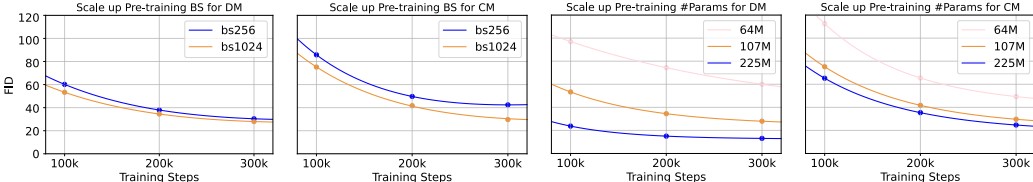

Figure 3: Performance in downstream tasks scales with pre-training compute budget. DM, CM: fine-tune under the settings of diffusion or consistency model.

counterpart. Besides, EPG-L/16 outperforms Shortcut-XL/2 in one-step generation, while leaving a small performance gap to its 4-step generation result. Noticeably, IMM marginally surpasses our model by 0.77 FID but requires 11× more training compute, highlighting practical advantages of our method. We display the FID convergence of EPG-L/16 in Figure 7.

## 4.2 ABLATION STUDY

**Comparing with pixel-space baselines**. On ImageNet-224, we compare with several pixel-space baselines in training diffusion and consistency model. (1) REPA (Yu et al., 2025), (2) load pre-trained MoCo v3 (Chen et al., 2021) weights, (3) train from scratch, and (4) pre-train with original rRCM framework (Lei et al., 2025). We display FID results in Table 6. The model equipped with REPA yields the worst generation quality. Besides, The model loaded with pre-trained MoCo v3 weights demonstrate better results and training stability than the one trained from scratch. In comparison with the model pre-trained by rRCM, our EPG demonstrates consistently better generation performance in both diffusion and consistency training tasks, indicating the remarkable effectiveness of our improvements. Notably, our framework is much concise and does not introduce coupled hyper-parameters. We defer implementation details of the baseline methods to Appendix B.1.

**Scalability.** We study the scalability of our method on pre-training batch size and model parameters with results shown in Figure 3. Specifically, we compare downstream model performance by increasing the pre-training batch size from 256 to 1024 (first two columns). In addition, we study the impact of the model parameters by scaling the encoder from 64M to 107M and 225M (last two columns). The decoder is scaled accordingly during fine-tuning, resulting in complete models of 116M, 229M, and 540M parameters, respectively. As demonstrated, the generation performance on downstream tasks improves as the pre-training budget increases. This scalability positions our framework as a promising solution for high-resolution and multi-modal image generation, where training and sampling efficiency are critical.

## 5 RELATED WORK

**Training diffusion model in pixel-space** Training diffusion models directly in pixel space is notoriously challenging due to high computational costs and slow convergence on high-resolution data. Prior work addresses this through optimizing model architectures (Jabri et al., 2023), improving diffusion formulations (Hoogeboom et al., 2023; Kingma & Gao, 2023), and decomposed training stages with masked pre-training (Lei et al., 2023) or cascaded diffusion models (Ho et al., 2021). While these efforts enhance generation quality of pixel-space diffusion, they still lag behind latent-

space methods in both performance and efficiency. Our work bridges this gap, establishing a new state-of-the-art for pixel-space approaches utilizing similar training cost as latent-space baselines yet with fewer sampling steps.

**Accelerate diffusion model training.** On one hand, current works in diffusion models are focusing on accelerating training speed by incorporating additional representational supervision. However, as far as we know, all these efforts remain confined in the latent-space training formulation. Yu et al. (2025) aligns intermediate features of diffusion models with those from off-the-shelf SSL models. As follow up works, several variants (Leng et al., 2025; Yao et al., 2025) have been proposed. Leng et al. (2025) train both VAE and diffusion models in an end-to-end manner, achieving superior training efficiency yet it still relies on trained VAEs. In contrast, another line of works (Chu et al., 2025; Stoica et al., 2025; Wang & He, 2025) speed up model training by incorporating representation learning loss, without relying on external SSL models. The most related work to ours is USP (Chu et al., 2025), which contains a strong representation learning stage to accelerate the training of diffusion models. In contrast, our study focuses on the image space instead of latent space. In addition, Stoica et al. (2025) and Wang & He (2025) accelerate model training by incorporating contrastive loss, imposed either on predicted clean images or intermediate features.

**Few-step generative models.** Consistency models (Song & Dhariwal, 2023; Song et al., 2023; Lu & Song, 2025) enable single-step sampling by enforcing consistency across deterministic diffusion trajectories. While theoretically appealing, CMs face amplified training challenges due to sparse supervision at narrow noise intervals. Solutions like weight initialization from pretrained diffusion models (Geng et al., 2025) mitigate this but remain infeasible without access to trained diffusion models. Meanwhile, Shortcut model (Frans et al., 2025) and IMM (Zhou et al., 2025) provide insightful solutions for achieving few-step generation with solid theoretical innovations. In contrast, from an empirical perspective, we offer an effective solution for training one-shot generators on high-resolution images directly in pixel space.

## 6 DISCUSSION

**Conclusion.** We present a two-stage training framework for training diffusion and consistency models in pixel space. It successfully bridges the performance and efficiency gap between pixel-space trainings and the latent-space approaches, reaching SOTA performance on ImageNet dataset. By identifying the semantic roles of encoder and decoder in diffusion-based generative models, we provide a practical pathway for efficiently training the generative models. We hope it will motivate future efforts to improve generation quality with well-established insights from the traditional visual domain.

**Advantages over VAE-based methods/LDM framework.** Our two-stage training framework fundamentally challenges the dominant LDM paradigm. It avoids the training complexity and performance bottleneck rooted in the VAE-based LDM framework, lowering the barrier to adapt generative models to new and unseen data. Besides, the pre-training is significantly efficient than VAE and the pre-trained model could immediately benefit the downstream generative model training (see Figure 7). As reported in Table 5, our diffusion model surpasses DiT in both performance and efficiency by a large margin. Moreover, by increasing model patch size proportionally to image resolutions (e.g. 16x16 on ImageNet-256 and 32x32 on ImageNet-512), our method benefits from significant training efficiency across resolutions in terms of both GFLOPs and training time.

**Limitations.** Our EPGs currently under-perform leading latent-space methods like REPA and RAE. However, the performance gap can be largely explained by a significant disparity in training compute. Our experiments in Table 1 support this, showing that the performance is limited by the overall training budget, not by a fundamental limitation of our approach. This finding positions our method as a highly scalable and promising approach for training pixel-space diffusion-based generative models. A natural next step, which we leave for future work, is to leverage powerful off-the-shelf backbones to accelerate this scaling and further close the performance gap.

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

## A    THE USE OF LARGE LANGUAGE MODELS (LLMS)

To enhance the clarity and readability of this work, we utilized DeepSeek[‡] for language refinement in several sections (e.g., abstract and introduction) of the manuscript. The tool's suggestions were reviewed and edited to ensure technical accuracy and alignment with the paper's scientific content.

## B    IMPLEMENTATION DETAILS

### B.1    BASELINE METHODS

We introduce the implementation details for baseline methods listed in Table 6.

**MoCo v3.** Utilizing pre-trained MoCo v3 ViT-B model (pre-trained for 300 epochs), we attach an additional decoder comprising 12 ViT layers and fine-tune the complete model end-to-end. Note that we also add residual connections between the encoder and decoder, and use adaLN-Zero in the decoder. Similar to our Base/16 model, the complete model contains 229M parameters. To enable class-conditional denoising training, we prepend an extra zero embedding to the encoder's original positional embedding. Besides, we concatenate the class and timestep embeddings with the noisy image tokens and feed them as input tokens to the model.

**REPA in pixel space.** We use the official implementation and train SiT-L/16 with the default hyper-parameter settings.

**rRCM.** We use the official checkpoint of pre-trained rRCM encoder. It has the same structure as our Base model while being pre-trained with 4096 batch size for 600k steps.

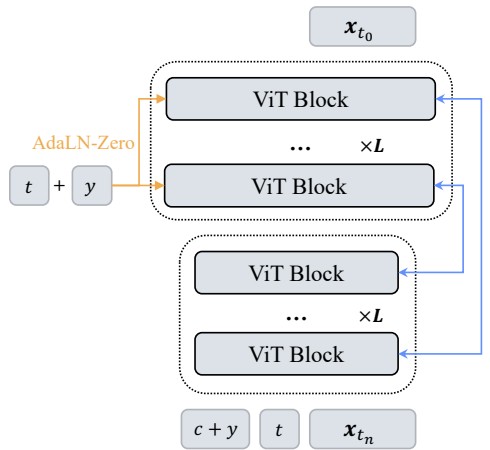

Figure 4: The network architecture of EPG utilized in downstream generative tasks.

Table 7: Network configurations of EPG. Encoder and decoder settings are separated by comma.

| Model | Blocks | Dim | Heads | Params |
|---|---|---|---|---|
| Small | 6, 6 | 768, 768 | 12, 12 | 116M |
| Base | 12, 12 | 768, 768 | 12, 12 | 229M |
| Large | 16, 16 | 1024, 1024 | 16, 16 | 540M |
| XL | 12, 12 | 768, 1584 | 12, 22 | 583M |
| XXL | 12, 12 | 768, 1920 | 12, 16 | 789M |
| G | 12, 12 | 768, 2688 | 12, 21 | 1391M |

Table 8: Network configurations of encoder in pre-training, named as Representation Consistency Model (**RCM**) for clarity.

| Model | Blocks | Dim | Heads | Params |
|---|---|---|---|---|
| RCM-S | 6 | 768 | 12 | 64M |
| RCM-B | 12 | 768 | 12 | 106M |
| RCM-L | 16 | 1024 | 16 | 225M |

### B.2    IMPLEMENTATION DETAILS

**Data pre-processing.** We pre-process ImageNet dataset by center-cropping images into $256 \times 256$ or $512 \times 512$ and save images in lossless PNG format.

**Training settings.**    We adopt the definition of SDEs and the time discretization strategy from EDM (Karras et al., 2022), as we discussed in Section 2. The time horizon is divided into $N$ non-overlapping intervals, where each interval endpoint $n$ is mapped to discretized time step (or noise level) via the following equation

$$t_n = (\sigma_{max}^{1/\rho} + \frac{n}{N-1}(\sigma_{min}^{1/\rho} - \sigma_{max}^{1/\rho}))^\rho, \quad \text{with} \quad n \in [0, N-1]. \tag{10}$$

---

[‡]https://www.deepseek.com

Table 9: Our pre-training and fine-tuning diffusion settings. †: In denoising training, we first sample $t$ from a continuous lognormal distribution and then map it to the nearest discrete value $t_n$. ‡: the denoiser predicts clean image $x$ at different noise levels.

| | Pre-training | Denoising Training | Consistency Training |
|---|---|---|---|
| Total discretization steps N | from 20 to 1280 | 1280 | $\infty$ (Continuous) |
| N or $p(r/t)$ | follow iCT (Song & Dhariwal, 2023) | Constant | follow ECT (Geng et al., 2025) |
| Weighting $\lambda(t)$ | 1.0 | $\frac{1}{t_n - t_{n-1}}$ | $\frac{1}{t-r}$ |
| Preconditioning | $1/\sqrt{t_n^2 + \sigma_{data}^2}$ | $1/\sqrt{t_n^2 + \sigma_{data}^2}$ | $1/\sqrt{t^2 + \sigma_{data}^2}$ |
| Time condition | $c(t) = 1000 * \frac{1}{4}\ln t$ | $c(t) = 1000 * \frac{1}{4}\ln t$ | $c(t) = 1000 * \frac{1}{4}\ln t$ |
| Noise Sampling | $n \sim \mathcal{U}(0, N)$ | $t_n \sim \mathrm{lognormal}(-1.2, 1.6)^\dagger$ | $t \sim \mathrm{lognormal}(-0.4, 1.6),\, r \sim p(r|t)$ |
| Training objective | Equation 8 | Equation 1 | Equation 5 + Equation 9 |
| Model parameterization | - | $x$-prediction‡ | Equation 4 |
| Shifting of noise level (Hoogeboom et al., 2023) | $image\_size$/64 | $image\_size$/64 | $image\_size$/64 |

Here, $\sigma_{max} = 80, \sigma_{min} = 0.002, \rho = 7$, $N$ is total discretization step. To regularize model input, we also employ the preconditioning $\frac{x_t}{\sigma_{data}+t^2}$ from EDM with $\sigma_{data} = 0.5$.

We consider our pre-training as a combination of representation learning and consistency training (Song & Dhariwal, 2023). Therefore, hyper-parameters for representation learning are grounded in SSL best practices: we use MoCo v3's augmentation strategies (Chen et al., 2021) for contrastive loss computation, set the EMA coefficient to 0.99 for the momentum encoder $E_{\theta^-}$, and adopt a linear learning rate schedule scaled to batch size (6e-4 for 1024 batch size). We use AdamW optimizer with default beta values. Built on top of these, we determine hyper-parameters of the representation consistency loss in equation 8. In specific, we adopt the time discretization annealing schedule from iCT (Song & Dhariwal, 2023), and implement a temperature schedule linearly interpolated between $\tau_1 = 0.1$ and $\tau_2 = 0.2$. See Section C for ablation on this design.

After pre-training, we fine-tune model under diffusion/consistency model training configurations. We utilize $x$-prediction for diffusion model and the EDM formulation (Equation 4) for consistency model. We also scale the noise level according to $image\_size$ of training data. Detailed pre-training and fine-tuning hyper-parameters are displayed in Table 10 and 11. For clarity, following (Lei et al., 2025), we name the pre-trained model as Representation Consistency Model (**RCM**) in our discussions below. We display the RCM model configurations in Table 8.

### B.3 NETWORK CONFIGURATION.

We list our model configurations in Table 7, and network structure in Figure 4. We use the Small and Base model in our ablation studies.

### B.4 COMPUTATIONAL COST

Utilizing 8×H200 GPUs, the pre-training takes 57 hours for RCM-B/16, 100 hours for RCM-L/16 on ImageNet-256, and 111 hours for RCM-L/32 on ImageNet-512. For the denoising task, training on ImageNet-256 takes 139 hours for EPG-XL/16, 160 hours for EPG-XXL/16, and 267 hours for EPG-G/16 (on 16×H200 GPUs). On ImageNet-512, training the EPG-L/32 model takes 100 hours. The EPG-L/16 consistency training on ImageNet-256 takes 156 hours.

## C ADDITIONAL ABLATION STUDY

In this section, we conduct ablation studies on the temperature value and training objectives of pre-training, and auxiliary loss in consistency fine-tuning. Besides, given pre-trained encoder, we also study how the flexibility of updating its parameters impacts downstream performance by controlling its learning rate. By default, We experiment with EPG-B/16 and pre-train the encoder for 300K steps. Subsequently, we fine-tune downstream diffusion models for 100K steps with 1024 batch size, and consistency models for 200K steps with 256 batch size.

Table 10: Hyper-parameters used in Pre-training.

|  | RCM-S, RCM-B | RCM-L |
|---|---|---|
| Bs | 1024 | |
| Training steps | 600k | |
| Lr | 6e-4, Cosine Decay | |
| Optimizer | AdamW(0.9, 0.999) | |
| Wd | 0.03 | 0.05 |
| $\tau$ for Rep consistency loss | linear interpolation with annealing $\tau_1 = 0.1, \tau_2 = 0.2$ | |
| $\tau$ for contrastive loss | 0.2 | |
| EMA of momentum encoder $E_{\theta^-}$ | 0.99 | |
| Diffusion settings | see Table 9 | |
| Data augmentation | Follow MoCo v3 (Chen et al., 2021) | |

Table 11: Hyper-parameters in Fine-tuning. We use step-wisely decayed learning rate when fine-tuning consistency model: in the starting 400k steps, lr=1e-4, from 400k to 500k, lr=3e-5, and lr=8e-6 thereafter. [†]: the dropout is employed after each Attention block in decoder.

|  | Denoising Training (Table 1) | Denoising Training (Table 2) | Consistency Training (Table 4) |
|---|---|---|---|
| Dataset | ImageNet-256 | ImageNet-512 | ImageNet-256 |
| Bs | 1024 | 1024 | 1024 |
| Training steps | 1000/2000K | 1000K | 700K |
| Lr | 1e-4 | 1e-4 | step-wisely |
| Grad clip | 0.5 | 0.5 | - |
| Optimizer | AdamW(0.9, 0.999) | AdamW(0.9, 0.999) | AdamW(0.9, 0.99) |
| Wd | 0.01 | 0.01 | 0.01 |
| Dropout | - | - | $0.5^{\dagger}$ |
| Patch size | 16×16 | 32×32 | 16×16 |
| Diffusion settings | See Table 9 | See Table 9 | See Table 9 |
| EMA value | 0.9999 | 0.9999 | 0.9999 |
| Data augmentation | Horizontal Flip | Horizontal Flip | Horizontal Flip |

**Impact of auxiliary loss in training consistency model.** In Table 13a, we compare consistency models trained without auxiliary loss ( equation 9) or without loading RCM weights. We use pre-trained weights from Table 1 at 600K step.

**Impact of temperature value.** We compare different temperature settings for representation consistency loss (1) No representation consistency loss (2) $\tau_1 = \tau_2 = 0.1$: fixed temperature 0.1, (3) $\tau_2 = 0.5$: linear interpolation with $\tau_2 = 0.5$, and (4) Ours: linear interpolation with $\tau_2 = 0.2$. We display results in 13b.

**Impact of pre-training loss terms.** In Table 13c, we conducted a detailed ablation study to isolate the contribution of each loss term to the final downstream generation performance. We observe that removing the contrastive loss causes performance to degrade severely, even more so than training from scratch. This is because the encoder fails to learn effective visual semantics, ultimately leading

Table 12: Ablation studies on auxiliary loss, temperature value and pre-training loss terms.

| Model | CM |
|---|---|
| Scratch | NaN |
| w/o aux | 129.16 |
| w/o RCM | 65.05 |
| EPG-B/16 | 36.75 |

(a) Auxiliary loss.

| Model | DM | CM |
|---|---|---|
| w/o consistency | 58.18 | 56.41 |
| $\tau_1 = \tau_2 = 0.1$ | 52.79 | 46.44 |
| $\tau_1 = 0.1, \tau_2 = 0.5$ | 56.22 | 45.69 |
| $\tau_1 = 0.1, \tau_2 = 0.2$ | 52.67 | 46.99 |

(b) Temperature value of RCL.

| Model | DM | CM |
|---|---|---|
| EPG-B/16 | 52.67 | 46.99 |
| w/o consistency | 58.18 | 56.41 |
| w/o constrast | 115.48 | 117.63 |
| Scratch | 77.25 | NaN |

(c) Pre-training loss term.

to collapsed representation across different noise levels. In contrast, a model trained from scratch is not constrained by this poor initialization, making it easier to optimize.

**Impact of the flexibility of encoder**. We use pre-trained RCM weights from Table 1 at 600K step and adopt the XL architecture during fine-tuning. Specifically, we compare three different settings: (1) *Frozen*: the pre-trained encoder is frozen and only decoder parameters are updated. (2) *Lrd=0.5*: we apply the layer-wise learning rate decay to encoder, where shallow layers (e.g. patch embedding layer) have smaller learning rate while deeper layers (e.g. last ViT block) have larger learning rate. (3) *Default*: default setting in our experiments, where the encoder is trained alongside decoder with a constant learning rate. For reference, we also present the FID of diffusion model trained from scratch. The results are displayed in Table 13.

The performance improves as encoder parameters are updated with greater flexibility. This result indicates that the pixel reconstruction inherent in denoising requires the encoder to supply low-level information, a capability it must adapt from its initial pre-training on high-level semantics.

## D    DETAILED QUALITATIVE RESULTS

**Training dynamics of downstream models.** We display the FID scores of downstream models with respect to training steps in Figure 7.

**Visualization of diffusion model encoder features.** To provide an intuitive comparison between our EPG diffusion variant and the one trained from scratch. We visualize output of their encoder features respectively in Figure 5 and 6. The results showcase the encoder features from our diffusion model yields stronger semantic structure.

## E    SEMANTICS LEARNED BY RCM ON NOISY SAMPLES.

To demonstrate that the RCM encodes noisy samples into meaningful representations, we compute the per-channel standard deviation of the $\ell_2$-normalized encoder outputs averaged across 1000 samples at various time steps. For RCM, the channel-wise std approximates $\frac{1}{\sqrt{d}}$, where $d$ is the feature dimension. In contrast, the encoder trained solely with contrastive loss exhibits near-zero std across channels at varying noise levels, indicating a representation collapse, as noted in prior work (Chen & He, 2020).

## F    QUALITATIVE RESULTS

We display more qualitative results in figures 9 10 11 12 13 14 15 16 17 18. Images are downsampled and compressed to optimize storage efficiency.

Noise decrease

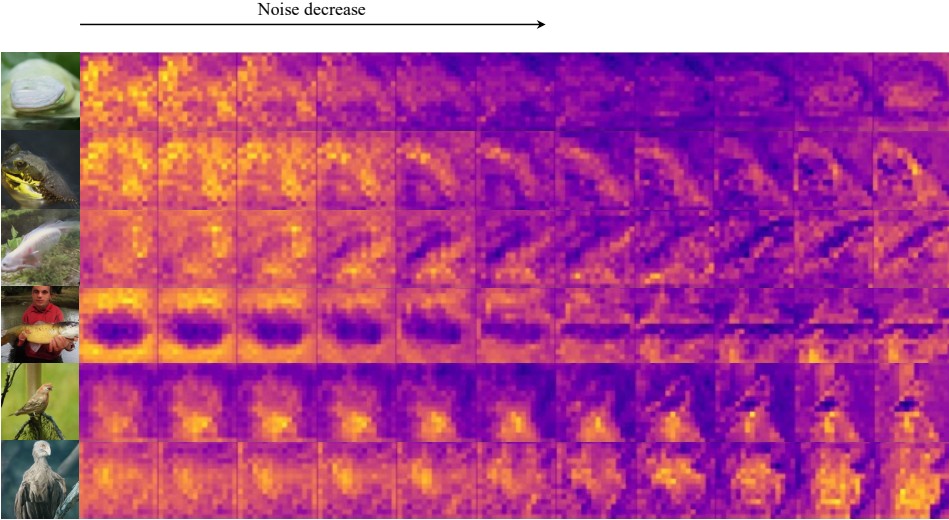

Figure 5: Visualizing encoder features of EPG-XL diffusion variant.

Noise decrease

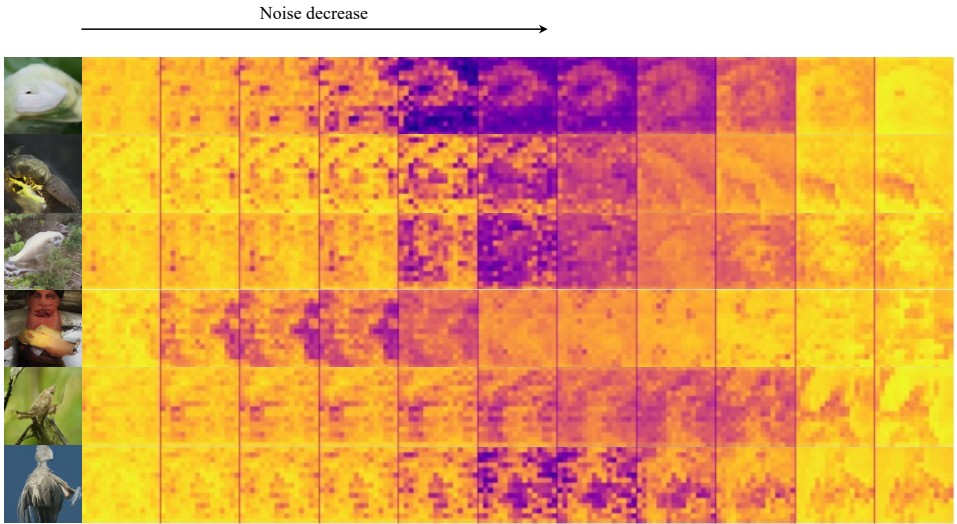

Figure 6: Visualizing encoder features of diffusion model trained from scratch.

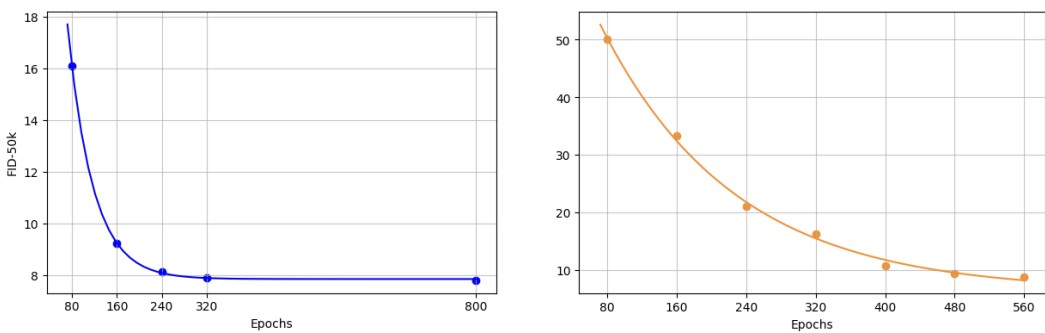

Figure 7: Training progress of downstream generative models: **(Left)** EPG-XL diffusion variant, **(Right)** EPG-L consistency variant.

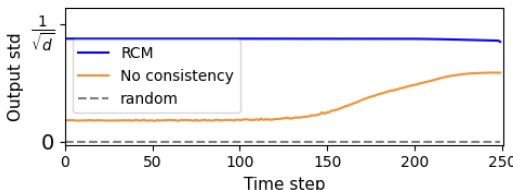

Figure 8: Avg per-channel std of RCM's outputs at varying time steps.

Table 13: Trade-off between the flexibility of and faithfulness to encoder representation.

| Encoder | FID |
|---------|-------|
| **Default** | **12.46** |
| Lrd=0.5 | 18.56 |
| Frozen | 22.44 |

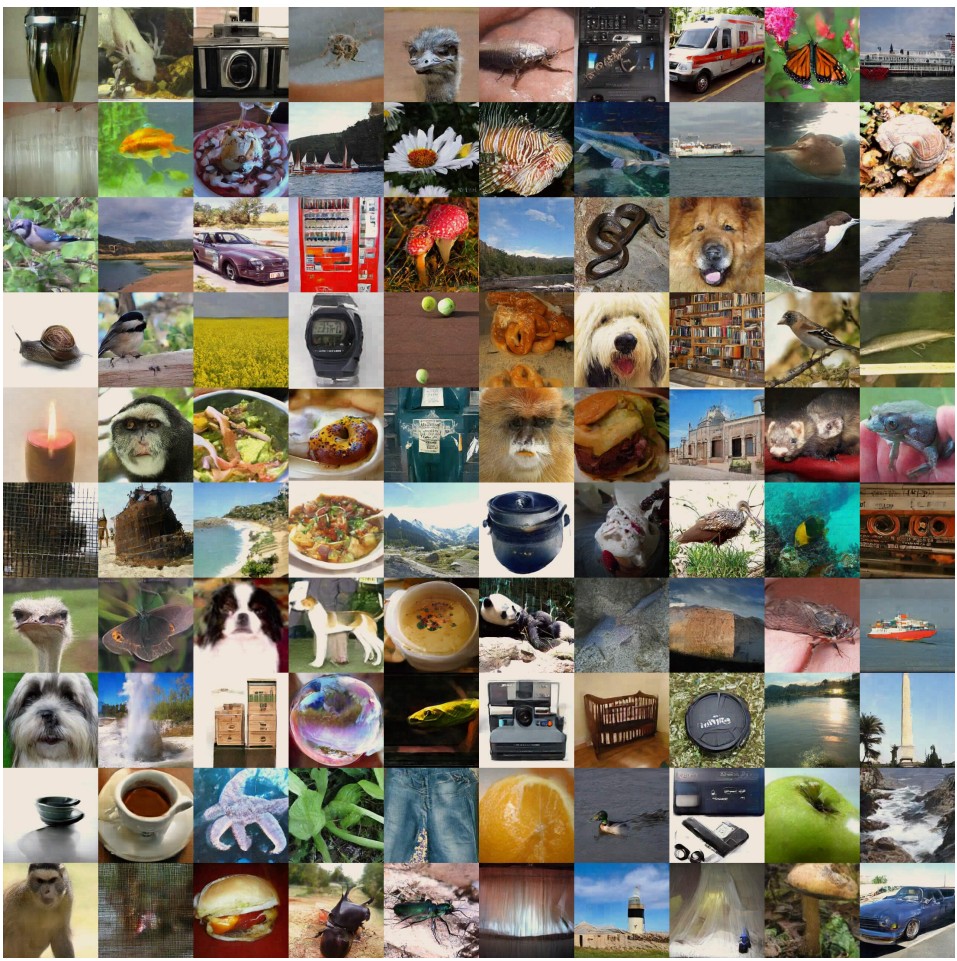

Figure 9: Images generated by EPG-L via one-step sampling.

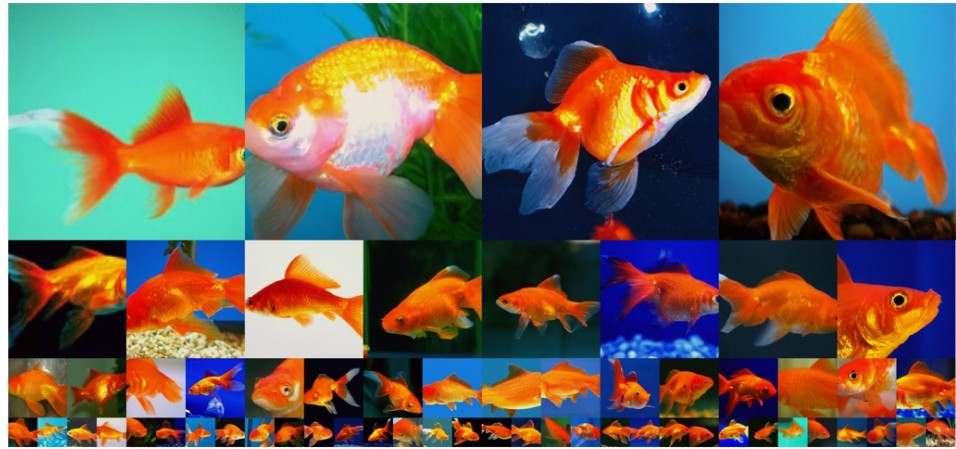

Figure 10: Uncurated samples generated by EPG-XL. 32-step Heun ODE sampler, classifier-free guidance 4.5, class 1.

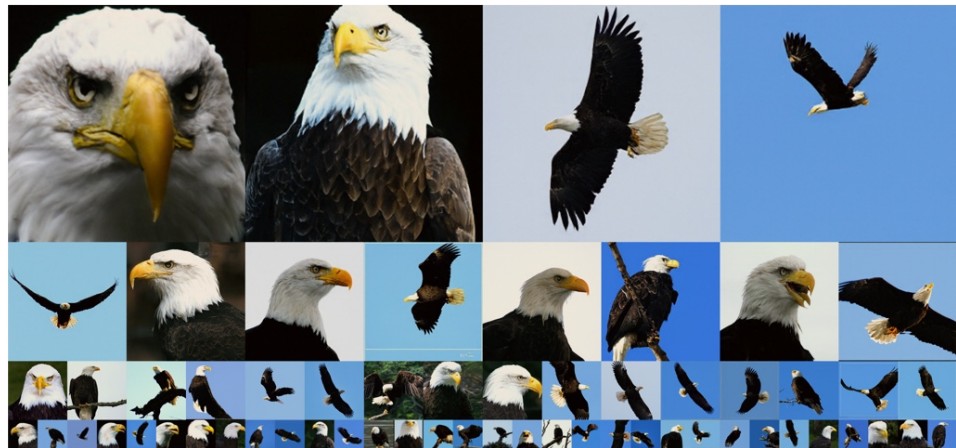

Figure 11: Uncurated samples generated by EPG-XL. 32-step Heun ODE sampler, classifier-free guidance 4.5, class 22.

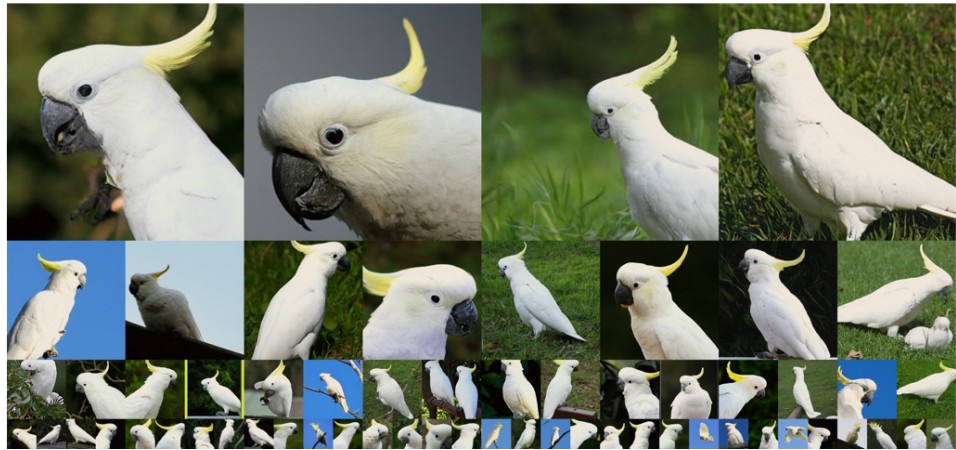

Figure 12: Uncurated samples generated by EPG-XL. 32-step Heun ODE sampler, classifier-free guidance 2.5, class 89.

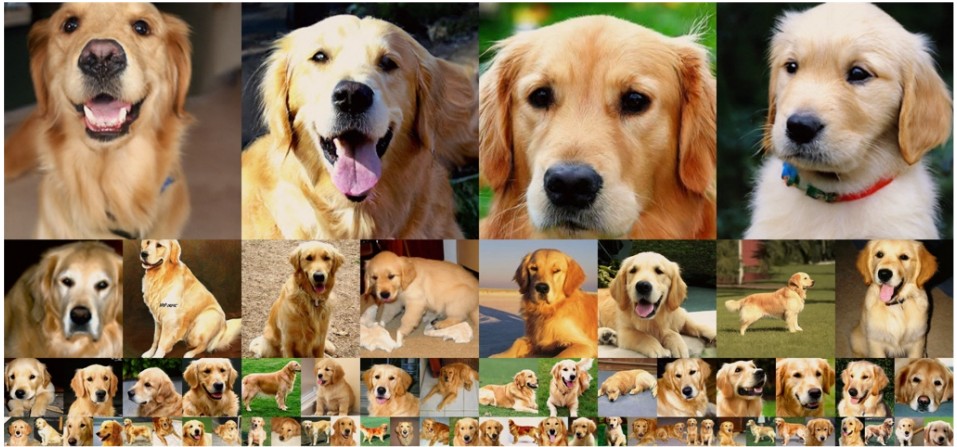

Figure 13: Uncurated samples generated by EPG-XL. 32-step Heun ODE sampler, classifier-free guidance 4.5, class 207.

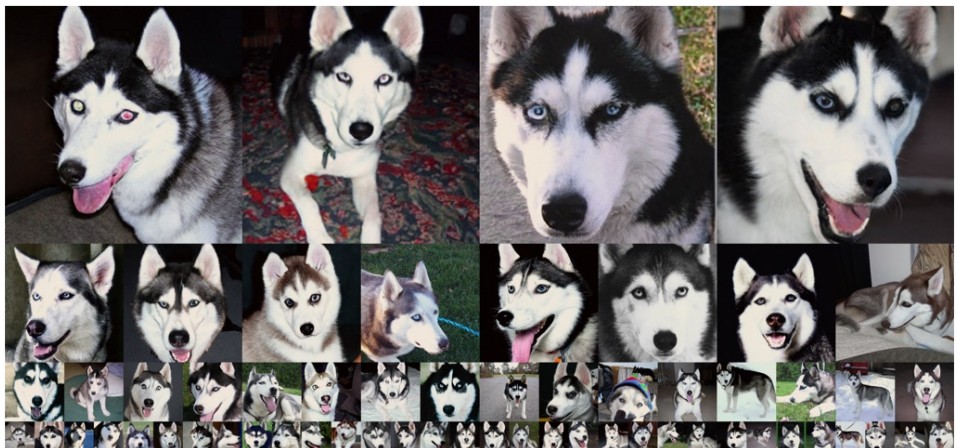

Figure 14: Uncurated samples generated by EPG-XL. 32-step Heun ODE sampler, classifier-free guidance 4.5, class 250.

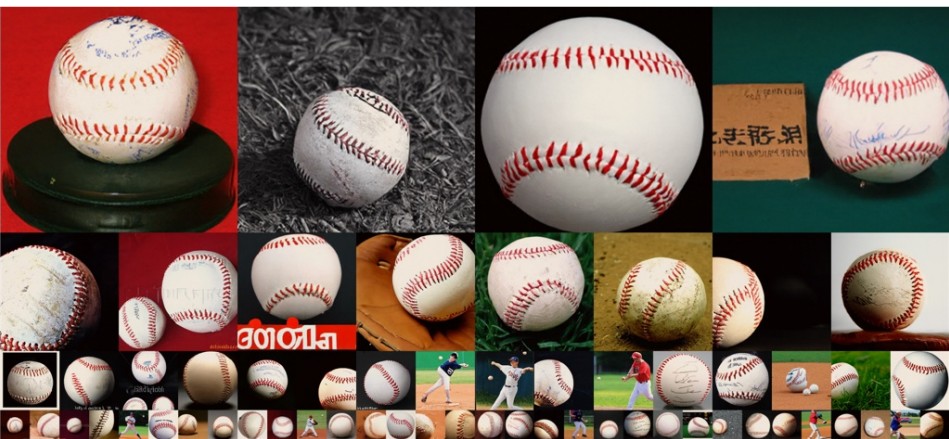

Figure 15: Uncurated samples generated by EPG-XL. 32-step Heun ODE sampler, classifier-free guidance 4.5, class 429.

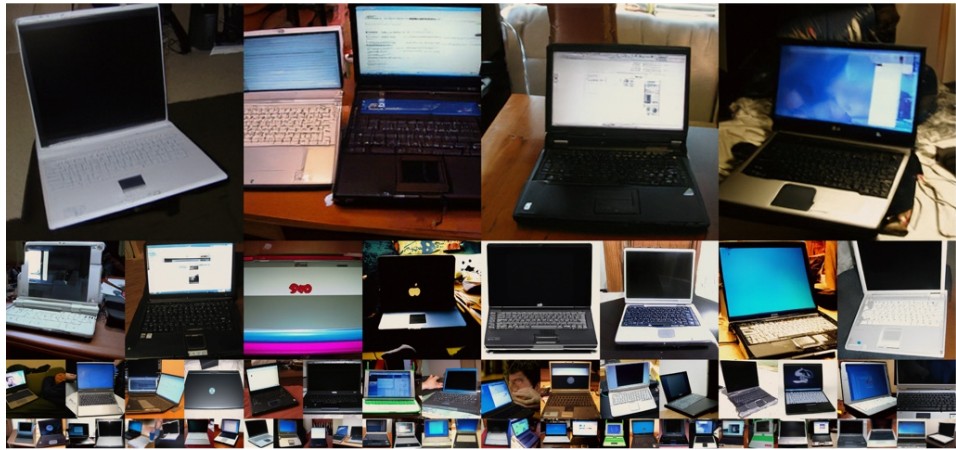

Figure 16: Uncurated samples generated by EPG-XL. 32-step Heun ODE sampler, classifier-free guidance 4.5, class 620.

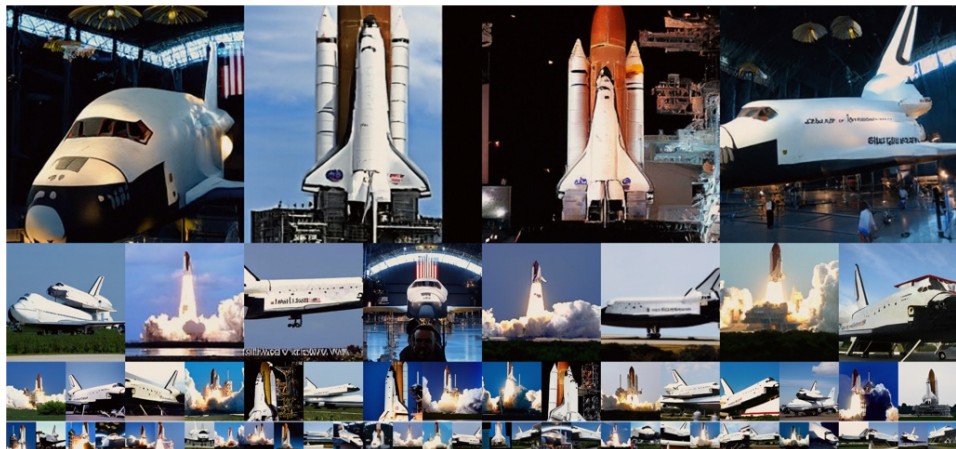

Figure 17: Uncurated samples generated by EPG-XL. 32-step Heun ODE sampler, classifier-free guidance 4.5, class 812.

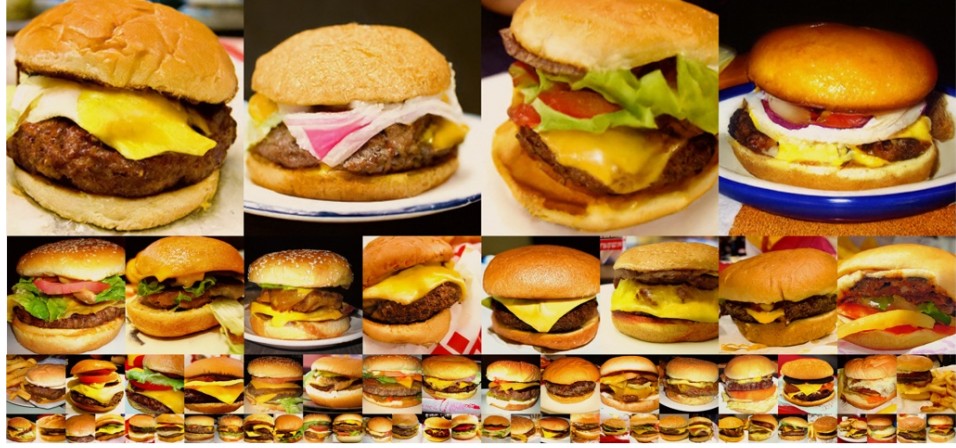

Figure 18: Uncurated samples generated by EPG-XL. 32-step Heun ODE sampler, classifier-free guidance 3.5, class 933.

