# OpenReview forum: "There is No VAE: End-to-End Pixel-Space Generative Modeling via Self-Supervised Pre-Training"
_ICLR.cc/2026/Conference — ICLR 2026 Poster_

### Official Review · Reviewer_jexG · 2025-10-18

**Soundness:** 3
**Presentation:** 2
**Contribution:** 4
**Rating:** 6
**Confidence:** 3

**Summary:**

This paper proposes a novel two-stage training framework for pixel-space consistency models. In the first stage, a portion of the generative model, referred to as the encoder, is pre-trained using self-supervised learning (SSL) methods, providing beneficial initialization. In the second stage, the entire model is fine-tuned with an additional network component, termed the decoder. This approach eliminates the need for latent diffusion training and achieves superior generation performance compared to existing end-to-end pixel-space generative models.

**Strengths:**

- The motivation is clearly articulated. In particular, the introduction effectively positions the work, helping readers understand its aims and benefits.
- The proposed method does not require external models, which may enhance its applicability to other domains.
- The overall training pipeline is presented clearly.
- The empirical results are impressive, and the evaluation is extensive.

**Weaknesses:**

[Major comments]
- Although two types of contrastive loss are proposed for the pre-training stage (Equation (8)), a more in-depth analysis of how each contributes to performance improvement (rather than reporting only FID scores) would be beneficial.
- It is unclear why $(\cdot)^-$ is applied to the first term and $sg$ to the second term in Equation (8). Could you provide further insight into this design choice?
- The fine-tuning stage lacks clarity, as $\theta'$ is not defined.

[Notation errors] There are several notation errors and unclear formulations:
- $\omega$ in the caption of Figure 2 should be $\theta$?
- The definitions of $(\cdot)^-$ in Lines 177 and 193 are inconsistent.
- $\tau_2(t)$ in Line 274 should be $\tau$?
- What is $\theta'$ in Equation (9)?
- In Equation (9), does $E_{\theta'}$ include the projection used in the pre-training stage?

[Minor]
- The reference for VAE cites the wrong publication year.

**Questions:**

- Have you experimented with performing the second-stage training while keeping the encoder, pre-trained in the first stage, frozen? Readers may be interested in understanding the trade-off between the flexibility of $f_\theta$ and the faithfulness to the representations learned in stage 1.
- How would the intermediate features obtained by $E_\theta$ differ if both $E_\theta$ and $D_\theta$ were trained in a single stage without the contrastive loss? If possible, visualizing this difference would be of interest to readers.

**Details Of Ethics Concerns:**

Potential ethical concerns are not discussed.

---

> ### Author Response · Authors · 2025-11-22
>
> We sincerely thank you for your detailed reviews and constructive suggestions to our presentations and experiments.
>
> **W.1: in-depth analysis of how each loss term contributes to performance improvement**
> We conducted a detailed ablation study to isolate the contribution of each loss term to the final downstream generation performance with results listed below (also updated in our submission, see Table 10.c in Appendix).  Removing the consistency loss degrades multi-step and few-step generation, while removing the contrastive loss causes performance to degrade severely, even more so than training from scratch.  We argue this is because the encoder fails to learn effective visual semantics, ultimately leading to collapsed representation across different noise levels.  In contrast, a model trained from scratch is not constrained by this poor initialization, making it easier to optimize.
>
> DM: FID-10k of diffusion model, CM: FID-50k of consistency model.
> | Model | DM | CM |
> | --- | --- | --- |
> | Default | 52.67 | 46.99 |
> | w/o consistency | 58.18 | 56.41 |
> | w/o contrast | 115.48 | 117.63 |
> | Scratch | 77.25 | NaN |
>
> **W.2: Parameter update rule for two target models in pre-training**
> During pre-training, we utilize two separate target models, each corresponds to a speicifc learning branch.
> - In semantics learning branch (compute contrastive loss in Eq.8), we use a slow EMA update (0.99 in our experiments) for the target model. The rationale is to prevent training collapse and provide a stable target for learning visual semantics.  This approach is standard in self-supervised literature, and we refer you to SimCLR[1] and MoCo[2,3] for a detailed discussion.
> - In semantic alignment branch (compute representation consistency loss in Eq.8), we use stop gradient operation for the target model. This means the target model is effectively frozen and shares the same parameters as the online encoder, which is updated with gradients. This design is grounded in the theoretical proofs of consistency models, where a parameter mismatch would lead to an invalid training objective and remove any convergence guarantee (see Section 3.2 in  iCT[4]).
>
> **W.3: Clarification on fine-tuning stage**
> We have restructured the fine-tuning section to separately and clearly describe the training procedures for the diffusion model and the consistency model, explicitly stating their respective pixel-level objectives and clarifying the role of the auxiliary loss in the latter.
>
> **Notation errors and typos:**
> In our previous submission, we aim to denote $E_{\theta^{'}}$ as a frozen copy of pre-trained encoder, and use $E_{\theta^{'}}$ to compute auxiliary loss. Note that, throughout consistency model training, $E_{\theta^{'}}$ is frozen and won't be updated.  Besides, we drop the projector layer $L_\theta$ after pre-trainng. Therefore, we do not use $L_\theta$ during fine-tuning.
> We have clarified this with different notations to avoid potential misleading and corrected all typos (caption in Figure 2, definitions of $\theta^-$, $\tau_2$, $E_{\theta^{'}}$) in our updated submission. Thank you for your detailed advice.
>
> **Q.1: Trade-off between the flexibility of and faithfulness to the encoder representation.**
> We conducted additional experiments on diffusion models in the Appendix, Table 11. Specifically, we compare three different encoder settings, while all decoders are trained with our default procedure:
> (a) The encoder is frozen.
> (b) The encoder is trained with layer-wise learning rate decay, where the learning rate gradually increases from shallower to deeper layers.
> (c) Our default setting, where the encoder and decoder are trained end-to-end.
> As expected, the performance improves as the encoder parameters are updated more flexibly. This is because the denoising task requires additional low-level details for pixel reconstruction, which contrasts with our pre-training objective of learning and aligning high-level semantics via contrastive loss [CITE MoCov3, SimCLR].
>
> | Encoder | FID |
> | --- | --- |
> | Default | 12.46 |
> | Lrd=0.5 | 18.56 |
> | Frozen | 22.44 |
>
> **Q.2: Visualization of features from pre-trained and from-scratch encoder in diffusion model**.
> We added the visualization in Appendix, Figure 4 and 5.  As demonstrated, our diffusion model yields stronger semantic structure when comparing with the one trained from scratch.
>
> We hope our response addresses all your concerns. Please let us know if any concerns remain unaddressed; we are happy to discuss them.

---

> ### Author Response · Authors · 2025-11-22
>
> [1] Chen, Ting, et al. "A simple framework for contrastive learning of visual representations." International conference on machine learning. PmLR, 2020.
> [2] He, Kaiming, et al. "Momentum contrast for unsupervised visual representation learning." Proceedings of the IEEE/CVF conference on computer vision and pattern recognition. 2020.
> [3] Chen, Xinlei, et al. "Improved baselines with momentum contrastive learning." arXiv preprint arXiv:2003.04297 (2020).
> [4] Song, Yang, and Prafulla Dhariwal. "Improved techniques for training consistency models." arXiv preprint arXiv:2310.14189 (2023).

---

> > ### Comment · Reviewer_jexG · 2025-11-25
> >
> > Thank you for your detailed rebuttal. I have carefully reviewed all the comments and the updates made to the manuscript. The authors’ responses satisfactorily address my questions, particularly regarding the roles of the proposed loss term, the parameter update rule, and the advantages of end-to-end encoder–decoder training during the fine-tuning step. I support the acceptance of this paper and will maintain my initial score.

---

### Official Review · Reviewer_nxB1 · 2025-10-28

**Soundness:** 3
**Presentation:** 2
**Contribution:** 3
**Rating:** 6
**Confidence:** 4

**Summary:**

The paper proposes a two-stage training framework — pre-training (representation consistency + contrastive learning) to help the encoder learn semantic features, followed by end-to-end fine-tuning with a randomly initialized decoder. The framework can be applied to pixel-space diffusion and consistency models, achieving strong results on ImageNet-256.

The paper is clearly written and easy to follow. The motivation, methodology, and experimental evidence are all presented in a logical and convincing manner.

In terms of originality, while the approach does not introduce fundamentally new techniques, it leverages existing methods in a novel way to improve the performance of high-dimensional pixel-space generative models, which is meaningful and valuable.

The experiments are relatively comprehensive and well-designed.

**Strengths:**

Please refer to Summary.

**Weaknesses:**

My main concern lies in the comparison with the latest latent-space methods, particularly RAE [1], where the proposed approach still lags behind in performance (ImageNet-256 FID 2.04 vs. 1.51). Moreover, most of the baselines used in the paper are relatively outdated (especially the pixel-level models), and the reported improvements over them are not very substantial.

Compared to VAE-based approaches, the proposed two-stage training pipeline does not demonstrate a clear advantage. Notably, recent works such as REPA-E [2] have also shown that end-to-end joint training of VAEs and diffusion models is feasible and effective.

Overall, while the method is conceptually simple, it does not appear to be sufficiently effective or provide strong new insights.

[1] Diffusion Transformers with Representation Autoencoders
[2] REPA-E: Unlocking VAE for End-to-End Tuning with Latent Diffusion Transformers

**Questions:**

Please refer to Weaknesses.

---

> ### Author Response · Authors · 2025-11-22
>
> We appreciate your thoughtful feedback, which situates our work in relation to other methods in the literature.
>
> **W.1: Performance lags behind recent latent-space models like RAE**
> We would like to point out that models like RAE leverage powerful, externally pre-trained components, such as a DINOv2-B encoder trained on a massive 142-million-image dataset. Our work, in contrast, intentionally trains from scratch purely on the target dataset (ImageNet, ~1.3M images). This distinction in overall training budge accounts for a significant part of the performance gap.
> To demonstrate this, we conduct additional experiments by scaling up our model parameters (from 583M to 789M and 1391M) and training duration (from 800 epochs to 1600 epochs) on ImageNet-256. This effort improved our FID score significantly, from 2.04 to 1.87, and further to 1.70 (displayed in our general response above, and is added to Table 1 in our updated submission). This new result narrows the gap with SOTA models substantially, **proving that our EPG's performance is not a fundamental limitation but a function of the training budget**
> To demonstrate this with more context, we compare with another concurrent work SVG [1], which achieves a 1.92 FID despite also leveraging DINO. Notably, our scaled-up EPG model (1.87 FID) already outperforms this approach, and with further scaling reaches an even stronger 1.70 FID—all while being trained on ImageNet. This provides compelling evidence for the high potential and competitive strength of our VAE-free paradigm.
>
> **W.2: Pixel-space baseline methods and the performance improvement over them**
> We compare with latest and parallel pixel-space methods, including **PixNerd (2.15 FID), PixelFlow (1.98), and JiT[2] (1.82)**. Our model  achieves the best generation performance, surpassing them by a large margin, e.g. 1.70 v.s. 2.15, 1.98, and 1.82.
> JiT, a concurrent work that proposes latest end-to-end pixel-space diffusion model, achieves 1.82 FID with 2B parameters, while our model, with ~70% model parameters, reaches 1.70 FID. While we share similar goals and observations (e.g., x-prediction, linearly scaled patch size, etc.), we additionally unveils that semantic quality on noisy images is essential to diffusion models, e.g., in terms of convergence speed and scalability. The later can be empirically demonstrated by the performance gain when we increase model parameters from 789M to 1391M during fine-tuning.  We also display the training progress of our EPG models in Appendix Figure 6.
>
> **W.3: Advantage of our method over VAE and REPA-E**
> - Performance: When making a fair comparison against VAE-based models that are also trained from scratch (like DiT and SiT), our advantage becomes clear. With a comparable parameter count (789M v.s. 759M), Our EPG model further surpasses them by a considerable margin (1.87 FID vs. 2.06 for SiT and 2.27 for DiT)
> -  Simplicity: Furthermore, our paradigm avoids the complexities of training VAEs, which requires balancing compression and reconstruction quality, often with unstable adversarial objectives.  Additionally, our pre-training significantly accelerates downstream training: both the diffusion and consistency models converge rapidly (Appendix Figure 6). In contrast, a pre-trained VAE offers only marginal benefits. This is evidenced by models like DiT and SiT that are trained from scratch within a given latent space.
> Regarding REPA-E, while it also supports end-to-end training, it does so with a hybrid approach that combines modules from different backgrounds. Moreover, by training the VAE simultaneously with the diffusion model, it tightly couples their training dynamics. This complicates the optimization process and increases sensitivity to hyper-parameter tuning.
> In contrast, our method avoid these problems, successfully training a diffusion model on raw pixels by focusing on the fundamentals of the denoising learning process and directly addressing the underlying training challenges.
>
> **W.4: Lack insights**
> We gently refer you to our general response above where we discuss our contributions and insights in details.
>
> We hope our response addresses all your concerns. Please let us know if any concerns remain unaddressed; we are happy to discuss them.
>
> [1] Shi, Minglei, et al. "Latent Diffusion Model without Variational Autoencoder." arXiv preprint arXiv:2510.15301 (2025).
> [2] Li, Tianhong, and Kaiming He. "Back to Basics: Let Denoising Generative Models Denoise." arXiv preprint arXiv:2511.13720 (2025).

---

> ### Author Response · Authors · 2025-11-26
>
> Dear Reviewer,
>
> Thank you for your thoughtful and constructive feedback on our work. We have posted our response and a revised version of the paper, in which we have aimed to address all of your comments.
>
> We are happy to engage in further discussion if any questions or concerns remain. We appreciate your time and consideration in re-evaluating our work.
>
> Best regards,
> Authors

---

### Official Review · Reviewer_FWeF · 2025-10-31

**Soundness:** 4
**Presentation:** 4
**Contribution:** 4
**Rating:** 4
**Confidence:** 4

**Summary:**

This paper proposes a novel two-stage training framework (tentatively referred to as EPG) for advancing end-to-end pixel-space generative models, addressing the long-standing limitations of Variational Autoencoder (VAE)-dependent paradigms in the field of diffusion and consistency models. The framework decouples the complex generative task into self-supervised pre-training and end-to-end fine-tuning phases: in the first phase, the encoder is trained independently via contrastive loss and representation consistency loss to learn noise-robust visual features; in the second phase, the pre-trained encoder is concatenated with a randomly initialized decoder for end-to-end optimization targeting downstream generative tasks. Experiments on ImageNet-256 and ImageNet-512 datasets demonstrate competitive performance—achieving FID scores as low as 2.04 and 2.35 respectively with only 75 inference steps—and pioneering pixel-space consistency model training without VAE reliance, yielding 8.82 FID in single-step generation. This work bridges the efficiency gap between pixel-space and latent-space models while simplifying the training pipeline.

**Strengths:**

Paradigm Innovation: Breaking VAE Dependence
The paper makes a significant paradigm contribution by eliminating the need for VAEs in high-quality generative modeling—a critical limitation of mainstream approaches like Stable Diffusion and DiT . VAEs introduce inherent trade-offs between compression ratio and reconstruction quality, and require costly joint fine-tuning across domains . By decoupling representation learning from pixel reconstruction, the proposed two-stage framework resolves these issues: the self-supervised pre-training phase ensures robust feature extraction from noisy data, while the lightweight fine-tuning phase avoids the complexity of VAE optimization. This "de-VAE" design aligns with emerging trends in pixel-space generation and significantly lowers the barrier to adapting generative models to new domains.

**Weaknesses:**

Limited Evaluation on High-Resolution and Diverse Datasets
While the paper demonstrates results on ImageNet-256/512, it lacks validation on higher-resolution tasks (e.g., 1024×1024 FFHQ) where pixel-space models historically struggle . The choice of ImageNet (natural images) also raises questions about generalizability to other domains (e.g., medical imaging, satellite imagery) where VAE artifacts are particularly problematic. Furthermore, key video generation metrics like temporal consistency or FVD are absent, despite the paper hinting at extensibility to video—leaving uncertainty about whether the framework can capture spatiotemporal dependencies.

**Questions:**

What is the impact of ODE path sampling density on the representation consistency loss and final generation quality? Could you supplement ablations or visualizations to clarify this mechanism?

---

> ### Author Response · Authors · 2025-11-22
>
> Thank you for your constructive feedback.
>
> **Weakenss: Limited Evaluation on High-Resolution and Diverse Datasets**.
> We agree with the reviewer that demonstrating the scalability and generalizability of our framework to higher resolutions and diverse data domains is a valuable and exciting direction for future research. For this work, we focused on the ImageNet-256/512 dataset, as it is the most direct and widely-accepted benchmark for our primary contribution. For example, recent works like MeanFlow[1], JiT[2], RAE[3] also report their primary results on the ImageNet dataset.
> The central aim of this paper is to prove that a pixel-space generative model can efficiently achieve performance competitive with leading VAE-based models (like DiT[4], iCT[5]) on large-scale, complex benchmarks. We appreciate the reviewer's suggestion of domains like medical and satellite imaging. While training on these specific datasets is beyond the scope of this initial work, our success on the general and diverse ImageNet dataset serves as strong evidence for its potential applicability. Regarding video generation,  a full exploration of this modality constitutes a significant research project in itself and is outside the scope of our current paper, which is focused on image generation.
>
> **Questions: impact of ODE path sampling density**.
> We assume "ODE path sampling density" refers to the noise level sampling distribution, which is set to uniform distribution during pre-training (detailed in the Appendix, Table 6). This sampling distribution balances the model training on different noise levels. A large probability density indicates the model is more frequently trained on specific noise levels. Our choice of the uniform distribution was determined empirically to balance loss magnitude across different noise levels.
> Moreover, we use the PF ODE path instead of normal noise augmentation because it provides a structured noise schedule. This allows the model to gradually align the semantics of noisy samples with cleaner ones, particularly at high noise levels, leading to stronger performance. For a detailed comparison, we refer you to rRCM[6] (Appendix, Section F).
>
> We hope our response addresses all your concerns. Please let us know if any concerns remain unaddressed; we are happy to discuss them.
>
> [1] Geng, Zhengyang, et al. "Mean flows for one-step generative modeling." arXiv preprint arXiv:2505.13447 (2025).
> [2] Zheng, Boyang, et al. "Diffusion transformers with representation autoencoders." arXiv preprint arXiv:2510.11690 (2025).
> [3] Li, Tianhong, and Kaiming He. "Back to Basics: Let Denoising Generative Models Denoise." arXiv preprint arXiv:2511.13720 (2025).
> [4] Peebles, William, and Saining Xie. "Scalable diffusion models with transformers." Proceedings of the IEEE/CVF international conference on computer vision. 2023.
> [5] Song, Yang, and Prafulla Dhariwal. "Improved techniques for training consistency models." arXiv preprint arXiv:2310.14189 (2023).
> [6] Lei, Jiachen, et al. "Robust Representation Consistency Model via Contrastive Denoising." arXiv preprint arXiv:2501.13094 (2025).

---

> ### Author Response · Authors · 2025-11-26
>
> Dear reviewer,
>
> We have submitted our response to your review. We believe we have addressed all the points you raised and welcome any further discussion. We would be grateful if you would consider our response and the revisions in your updated assessment of our work.
>
> Best regards,
> Authors

---

### Official Review · Reviewer_bsaA · 2025-11-03

**Soundness:** 2
**Presentation:** 2
**Contribution:** 3
**Rating:** 4
**Confidence:** 3

**Summary:**

The authors propose a two-stage framework for (1) representation learning with consistency regularization, and (2) fine-tuning for the generative model. To make the optimization of the representation learning in the first stage work, the authors leverage EMA and stop gradients to form the training objectives. The fine-tuning is conducted using a diffusion model and extra regularization to ensure that the encoder remains structured and meaningful.

**Strengths:**

The authors carefully designed the training objectives for leveraging consistency loss in the training of the encoder.

The authors focus on important questions and give good justifications for their initiatives.

**Weaknesses:**

My main concern is that I am not sure if the paper can be claimed as a "pixel-based" generative model, since:

(1) The consistency regularization is conducted in the latent space.

(2) The generative model is also trained in the diffusion model. (Also, I do think the authors should specify how the generative model is trained in their diagram, as from the current one, you might think they are learning a generative model in the 1st stage of learning representations, which is not the case.

**Questions:**

Have you evaluated the encoder's performance?

How's the performance if you don't use the consistency regularization? Or simply using it without the contrastive objective.

---

> ### Author Response · Authors · 2025-11-22
>
> Thank you for your constructive feedback and for raising these important questions.
>
> **W: On the claim of being a "pixel-based" generative model:**
> Our model is fundamentally "pixel-based" because its core generative process operates directly on raw pixels, from input to output, without relying on a pre-trained, fixed tokenizer (like a VAE) to project data into a separate latent space. We believe the confusion arises from two points, which we clarify below:
> 1. The role of the latent space during pre-training:
> The reviewer correctly notes that our representation learning and consistency regularization is applied in a latent space. The crucial distinction is that this is the internal, learned latent space of our own encoder, not a fixed latent space from a separate, pre-trained model like in Latent Diffusion Models (LDMs).
> Our Method vs. LDM: LDMs are trained exclusively in the latent space of a frozen VAE. They cannot operate on pixels directly. In contrast, our framework trains the encoder of generative model from scratch and then fine-tunes the entire generative model end-to-end on pixels. The latent-space objective in our first stage is a pre-training strategy to intialize the encoder weights, analogous to self-supervised pre-training in computer vision. The final generative model, however, is a single, cohesive network that processes pixels.
> 2. The role of the encoder during fine-tuning:
> The generative model (diffusion/consistency) is primarily trained with a pixel-level objective while the consistency model is additionally trained with an auxiliary loss.
> -  For our diffusion model, the training uses the standard pixel-space denoising objective (Eq. 1). The pre-trained encoder simply provides a good weight initialization.
> - For our consistency model, we additionally use an auxiliary loss.  This is a regularization technique and Its purpose is to provide additional supervisory signals to stabilize and accelerate convergence, a known challenge when training consistency models [1,2,3]. The model itself still learns to map noise to pixels.  We presented ablation study in Appendix Table 10.a that demonstrates the contribution of both the pre-trained weights and this auxiliary loss, confirming it acts as a beneficial regularizer. For your convenience, we list the results below.
>
> | Model | FID |
> | --- | --- |
> | EPG | 36.75 |
> | w/o pre-train | 65.05 |
> | w/o aux | 129.16 |
> | Scratch | NaN |
>
> **Revisions to Improve Clarity:**
> Our original diagram and description may have conflated these points. To address this directly, we have significantly simplified our main framework diagram (Figure 2) to clearly distinguish the two stages. Besides, we also restructured the fine-tuning section to separately and clearly describe the training procedures for the diffusion model and the consistency model, explicitly stating their respective pixel-level objectives and clarifying the role of the auxiliary loss in the latter.

---

> ### Author Response · Authors · 2025-11-22
>
> **Q1: Evaluating the performance of pre-trained encoder**.
> Our pre-trained encoder is trained with contrastive loss to capture meaningful representation and consistency regularization to ensure the model outputs consistent and meaingful feautures given noisy inputs.  Therefore, we evaluate the pre-trained encoder from two key perspectives that directly correspond to its design goals: semantic quality, and semantic consistency under noise. We list the results below and have also added it to our updated submission (Appendix, Table 10.c )
> - Semantic quality: in self-supervised learning, a typical way to measure the quality of learned visual semantics is to evaluate the linear probing accuracy of trained encoder in image classification task. A strong performance here indicates a strong semantic quality.
> - Semantic consistency: we follow prior works RCG[4] and report the distribution distance, named as FD, between the features of clean images and pure noise, all while using our pre-trained encoder to extract features. The lower the score, the closer the learned features on noisy images are to clean images.
>
> | Model | LinP | FD |
> | --- | --- | --- |
> | Encoder in Table 1, IN-256 | 69.80 | 1.21 |
> | Encoder in Table 4, IN-256 | 72.25 | 0.82 |
> | Encoder in Table 2, IN-512 | 71.93 | 0.67 |
> | RCG | - | 0.48 |
>
> To establish a better reference point for the FD score, we cite the results from RCG[4]. In their work, they show that training a diffusion model to approximate features of a trained semantic encoder yields an FD score of 0.48. Our FD score strongly supports the semantic consistency of our pre-trained model.
>
> **Q2: Ablation of the two loss terms used during pre-training**.
> We conducted a detailed ablation study to isolate the contribution of each loss term to the final downstream generation performance with results listed below (also updated in our submission, see Table 10.c in Appendix).  Removing the consistency loss degrades multi-step and few-step generation, while removing the contrastive loss causes performance to degrade severely, even more so than training from scratch.  We argue this is because the encoder fails to learn effective visual semantics, ultimately leading to collapsed representation across different noise levels.  In contrast, a model trained from scratch is not constrained by this poor initialization, making it easier to optimize.
>
> DM: FID-10k of diffusion model, CM: FID-50k of consistency model.
> | Model | DM | CM |
> | --- | --- | --- |
> | Default | 52.67 | 46.99 |
> | w/o consistency | 58.18 | 56.41 |
> | w/o contrast | 115.48 | 117.63 |
> | Scratch | 77.25 | NaN |
>
> We hope our response addresses all your concerns. Please let us know if any concerns remain unaddressed; we are happy to discuss them.
>
> [1] Lu, Cheng, and Yang Song. "Simplifying, stabilizing and scaling continuous-time consistency models." arXiv preprint arXiv:2410.11081 (2024).
> [2] Geng, Zhengyang, et al. "Consistency models made easy." arXiv preprint arXiv:2406.14548 (2024).
> [3] Song, Yang, and Prafulla Dhariwal. "Improved techniques for training consistency models." arXiv preprint arXiv:2310.14189 (2023).
> [4] Li, Tianhong, Dina Katabi, and Kaiming He. "Return of unconditional generation: A self-supervised representation generation method." Advances in Neural Information Processing Systems 37 (2024): 125441-125468.

---

> ### Author Response · Authors · 2025-11-26
>
> Dear reviewer,
>
> Thank you again for your constructive feedback. We hope you will find that your concerns have been fully addressed. Please let us know if you have any follow-up questions; we would be happy to discuss them further.
>
> Best regards,
> Authors

---

### Author Response · Authors · 2025-11-22
**General Response to All Reviewers (2/2)**

Diffusion generation performance on ImageNet-512. We use 32 $\times$ 32 patch size. The result has been added to Table 2 in our updated submission.
| Model | FID | Epochs | Parameters | FLOPs |
| --- | --- | --- | --- | --- |
| DiT-XL/2 | 3.04 | 600 | 84M+675M | 1260+525 |
| SiT-XL/2 | 2.62 | 600 | 84M+675M | 1260+525 |
| SiD | 3.02 | 800 | 2.46B | - |
| PixNerd | 2.84 | 320 | 700M | 583 |
| VDM++ | 2.65 | 800 | 2.46B | - |
| EPG |  2.43 |  600 | 540M | 113 |
| EPG |  2.35 |  800 | 540M | 113 |

### 2. List of Revisions
Presentation
- Figure 2: We have significantly simplified our main framework diagram (Figure 2) to clearly distinguish the two stages. It now explicitly shows that the generative model is trained in a separate, second stage.
-  Section 3.3 fine-tuning: We restructured the fine-tuning section to separately and clearly describe the training procedures for the diffusion model and the consistency model
- We have corrected the typos pointed out by the reviewers.

Experiment
- In Table 1, added experiments on ImageNet-256 with scaled-up model sizes.
- In Table 2, added experiments on ImageNet-512
- In Appendix Table 10.c, added an ablation study to isolate the contribution of pre-training loss terms.
- In Appendix Table 11, added an ablation study on denoising training that constrains the encoder learning rate.
- In Appendix Figures 4 and 5, added a visualization of the our downstream diffusion model encoder features, comparing it to one trained from scratch.
- In Appendix Figure 6, added visualization of the FID of downstream generative models with respect to training steps.

---

### Author Response · Authors · 2025-11-22
**General Response to All Reviewers  (1/2)**

### 1. We would like to clarify the primary contributions/insights of our work:
- Insight 1: Diffusion-based generative models don't need tokenizers.

The LDM framework's reliance on a VAE tokenizer introduces significant issues. Firstly, training the VAE itself is difficult due to the need to balance compression with high-fidelity reconstruction. Besides, even when trained properly, the VAE often produces imperfect reconstructions for latents far from the training set. While pre-training the VAE on massive datasets can mitigate this, it will incur heavy computational costs prior to training generative models. Moreover, it induces a permanent performance bottleneck, as the generative model's ability to adapt to new data is always limited by the VAE's fixed capacity. We avoids the above issues  by successfully training diffusion-based generative models directly in pixel space.

While concurrent works, such as RAE[1], achieve impressive results, they did not resolve the long-standing issues associated with tokenizer-based generation. They follow the LDM framework while substituting the conventional VAE with a more powerful, pre-trained representation model (e.g., DINOv2) to act as a new, fixed tokenizer. However, this approach still constrains the generative model with its reliance on an external tokenizer; It does not fundamentally solve the bottleneck of the LDM framework but merely replace it.

Our work, however, takes a different path. We challenge the notion that a separate, pre-trained tokenizer is a prerequisite for training high-performing generative models. Our primary contribution is not to engineer a system that sets a new FID record, but to present a new, general-purpose training algorithm that **fundamentally challenges this dominant paradigm**.

- Insight 2: We empirically show that the roles of the encoder and decoder in diffusion-based generative models can be largely decomposed. This allows downstream models to benefit from the strong semantic representations (across different noise levels) provided by a pre-trained encoder. As we demonstrate below, a key benefit of this approach is significantly improved scalability.

To eliminate the tokenizer bottleneck, we hypothesize that the roles of the encoder and decoder can be decomposed, leading to separate training stages. Specifically, we assume that the encoder acts as a semantic learner on noisy images, and the decoder acts as a pixel reconstruction model.  Our two-stage framework validates this on ImageNet, showing that providing strong, pre-trained semantics across all noise levels from the start significantly benefits the downstream generative model. This also explains why common pixel-space diffusion and consistency models yield worse performance: their pixel-wise prediction loss is inefficient in learning task-specific semantics (at different noise levels), which is crucial for model convergence and final generation performance.

While JiT[2], **a recent work made public during the review process**, successfully trains diffusion models with a transformer architecture, our conclusion is still supported by its performance on ImageNet-256, which only improves marginally when scaling up model parameters, a directly consequence of inefficient semantic learning . To validate this, we conducted further experiments on ImageNet-256, presented below.
With 789M parameters, EPG surpasses the performance of JiT-H (953M) and JiT-G (2B). Besides, with extended training and parameters up to 1391M params, the generation performance further improves, reaching 1.70 FID (1.58 FID with class-balanced sampling). **These empirical results strongly indicate a promising direction for scaling up pixel-space diffusion models.**

Pixel diffusion generation performance on ImageNet-256.  $^\dagger$: adopt class-balanced sampling. The result has been added to Table 1 in our updated submission.
| Model | FID | Epochs | Parameters | FLOPs |
| --- | --- | --- | --- | --- |
| PixelFLow [3] | 1.98 | - | 677M | 2909 |
| PixNerd [4] | 2.15 | 160 | 700M | 134 |
| JiT-H$^\dagger$ | 1.86 | 600 | 953M | 182 |
| JiT-G$^\dagger$ | 1.82 | 600 | 2B | 383 |
| EPG |  2.04 |  800 | 583M | 128 |
| EPG$^\dagger$ |  1.81 |  600 | 789M | 176 |
| EPG |  1.70 |  1600 | 1391M  | 321 |
| EPG$^\dagger$ |  **1.58** |  1600 | 1391M  | 321 |

[1] Zheng, Boyang, et al. "Diffusion transformers with representation autoencoders." arXiv preprint arXiv:2510.11690 (2025).
[2] Li, Tianhong, and Kaiming He. "Back to Basics: Let Denoising Generative Models Denoise." arXiv preprint arXiv:2511.13720 (2025).

---

### Author Response · Authors · 2025-12-01
**A Brief Summary of Rebuttal**

Dear AC:

We are very grateful for your thorough review and the considerable effort you invested, particularly in light of the recent difficulties. We also appreciate the reviewers for their constructive feedback, which has helped us strengthen our work.

We have provided detailed responses to questions from all reviewers and have refined our paper accordingly. During the rebuttal period, Reviewer jexG has actively engaged in discussion with us and **confirmed that our responses and revisions satisfactorily address their concerns, leading them to support acceptance. We believe our detailed responses and comprehensive revisions have fully addressed the concerns raised by all reviewers.**

To facilitate your evaluation, we would like to summarize the contributions of our work, its strengths as confirmed by the reviewers, and the major revisions we have made.
### Contributions
We propose a novel two-stage training framework that first bridges the performance and efficiency (training and inference) gap between pixel- and latent-space diffusion-based generative models. Noticeably, our method achieves this **without relying on any external models, such as VAE or DINO.**  By successfully identifying the semantic roles of the encoder and decoder, we establish that training a diffusion model can be framed as a self-supervised learning problem, similar to training an image classifier.

In our experiments, our diffusion model EPG-G/16 reaches **an FID of 1.58 on ImageNet-256** and EPG-L/16 reaches 2.35 on ImageNet-512 with 75 number of function evaluations, setting a new SOTA on ImageNet-256 for VAE-free diffusion models.
For the first time, we close the gap with the dominant latent-space paradigm and achieve performance that surpasses leading latent diffusion counterparts. For example, **our EPG-XXL achieves 1.87 FID with ~33% overall training cost of DiT (2.27 FID)** (See Table 1 and 5 for more details).

Furthermore, in terms of both GFLOPs and training time, our model, built on Vision Transformer, maintains significant computational efficiency across different resolutions. This is achieved by fixing input token length through proportionally adjusting the patch size as image resolution increases, e.g., 16×16 on ImageNet-256 and 32×32 on ImageNet-512.

Meanwhile, our consistency model achieves **an impressive FID of 8.82 in a single sampling step, significantly surpassing its latent-space counterpart**. This marks the first time a consistency model has been successfully trained directly on ImageNet-256 without utilizing pre-trained VAEs or diffusion models.

The empirical results establish our training framework as an efficient, performant, and scalable solution for pixel-space generation.

### Strength confirmed by reviewers
As highlighted by the reviewers, our method:
- Presents a simple, effective, and well-motivated idea. [Reviewer 2, FWeF, Reviewer 3, nxB1, Reviewer 4, jexG]
- Addresses an important and timely problem, making a valuable contribution to the field of generative modeling. [Reviewer 2, FWeF, Reviewer 3, nxB1, Reviewer 4, jexG]
- Yields competitive/strong/impressive empirical results [Reviewer 2, FWeF, Reviewer 3, nxB1, Reviewer 4, jexG]

### Major revisions to our paper (highlighted in blue)
- Clarity and presentation
**(a)** Clarified our core insights and contributions in the introduction. `[Reviewer nxB1]`
**(b)** Redesigned the main framework diagram (Figure 2) to clearly distinguish the two training stages. `[Reviewers bsaA, jexG]`
**(c)** Restructured the fine-tuning section (section 3.3) to separately and clearly describe the training procedures for the diffusion model and the consistency model. `[Reviewers bsaA, jexG ]`
**(d)** Added a discussion section to highlight its advantages over VAE-based methods. `[Reviewer nxB1]`

- Experiments and visualizations
**(a)** Added extensive results on ImageNet-256 and -512, showcasing our method's state-of-the-art performance, scalability, and efficiency at multiple resolutions. `[Revewier nxB1]`
**(b)** Added performance and training efficiency comparison with DiT in Table 5 to further demonstrate our advantages over the VAE-based LDM framework. `[Reviewer nxB1]`
**(c)** Conducted new ablation studies on the contribution of each pre-training loss term (Table 11.c) and the flexibility of updating the pre-trained encoder (Table 12). `[Reviewers bsaA, jexG]`
**(d)** Added visualizations of the encoder features (Figure 4 and 5), demonstrating the benefits of our pre-training stage compared to training from scratch. `[Reviewer jexG]`
**(e)** Added visualizations of the training progress (Figure 6), showing the rapid convergence of our downstream model's FID score. `[Reviewer nxB1]`

Best regards,
Authors

---

> ### Author Response · Authors · 2025-12-02
>
> We would like to highlight our responses to the primary concerns raised by the reviewers:
>
> **1. Advantages over VAE-based methods/LDM framework**
>
> Our two-stage framework presents a compelling and more efficient alternative to the dominant Latent Diffusion Model (LDM) paradigm.
> - Firstly, It avoids the training complexity and performance bottleneck rooted in the VAE-based LDM framework, lowering the barrier to adapting generative models to new and unseen data.
> - Besides, our pre-training is significantly efficient than VAE, and the pre-trained model could immediately benefit the downstream generative model training (see Table 5 and Figure 6). As reported in Table 5, our diffusion model significantly surpasses DiT in both performance and overall training efficiency, all without relying on any external models (e.g., VAE, DINO).
> - Moreover, our framework scales efficiently to higher resolutions by maintaining a fixed number of input tokens. We achieve this by proportionally increasing the patch size with the image resolution (e.g., 16x16 for 256px, 32x32 for 512px), keeping computational cost (GFLOPs) and training time remarkably stable. In contrast, latent-space methods lack this flexibility. To maintain a fixed token count at higher resolutions, their VAEs would require a higher compression rate, which often leads to information loss and degraded downstream generation quality, undermining their efficiency at scale.
>
> **2. Comparing with latest and concurrent  pixel-space methods**
> We additionally compare our model against the latest and concurrent works, such as JiT, PixNerd, and PixelFlow. As detailed in Table 1 (also displayed the results in our General Response to All Reviewers (1/2)), our method consistently achieves the best generation performance. These results firmly establish our method as a leading and state-of-the-art solution within the pixel-space diffusion literature.

---

### Meta-Review · Area_Chair_wAU7 · 2025-12-15

**Summary:**

The paper proposes a novel two-stage training framework (Self-Supervised Pre-training followed by End-to-End Fine-tuning) designed to eliminate the reliance on pre-trained VAEs and tokenizers in pixel-space diffusion. Reviewers recognized the clear motivation of removing the "tokenizer bottleneck" and avoiding VAE compression artifacts. Primary concerns initially focused on the method's performance relative to state-of-the-art Latent Diffusion Models (like RAE and DiT), the semantic clarity of the pixel-space definition given the latent pre-training stage, and the limitation of experiments to ImageNet-256. During the rebuttal, the authors provided extensive new results, including scaling the model to achieve an FID of 1.58 on ImageNet-256 (surpassing concurrent pixel-space works) and adding ImageNet-512 results.

**Reviewer Concerns:**

**Reviewer Concerns Addressed:**

  - Performance Gap vs. Latent Models: Reviewer nxB1's concern that the method lagged behind recent latent models (e.g., RAE) was effectively addressed. The authors scaled up their model (from 583M to 1391M parameters) and training duration, achieving an FID of 1.58 (down from 2.04), which outperforms concurrent pixel-space methods like JiT and narrows the gap with SOTA latent models.

  - Clarification of Pixel-Space vs. Latent: Reviewer bsaA's concern regarding the terminology was addressed by clarifying that the "latent" stage refers to the internal representations of the encoder being trained from scratch, rather than a fixed, external VAE tokenizer.

  - Technical Formulation & Ablations: Reviewer jexG’s questions regarding the contribution of specific loss terms and encoder flexibility were resolved via new ablation studies (Table 10.c and Table 11), which isolated the impact of consistency vs. contrastive losses. Reviewer jexG explicitly confirmed these revisions were satisfactory.

**Outstanding Concerns:**

  - Reviewer FWeF suggested evaluation on diverse domains (medical, satellite). While the authors argued ImageNet is the standard benchmark for this class of generative models, the paper still relies solely on ImageNet, leaving the method's "generalizability" to non-natural images empirically untested in this specific manuscript.

**Reviewer Scores:**

**Reviewer jexG:** Initial Score: 6. This reviewer actively engaged in the rebuttal and explicitly stated they support acceptance and will maintain their score after the authors addressed their technical queries (guess: 6).

**Reviewer nxB1:** Initial Score: 6. This reviewer was concerned about the performance gap with RAE. Given the authors provided significant new results showing the model scales to beat concurrent works (JiT) and rivals DiT efficiency, it is highly likely their assessment would improve (guess: 8).

**Reviewer bsaA:**  Initial Score: 4. This reviewer’s main issue was the "pixel-based" claim and diagram clarity. The authors simplified the diagram and clarified the definitions. While the reviewer might still debate the terminology, the technical soundness was reinforced (guess: 6).

**Reviewer FWeF:** Initial Score: 4. This reviewer provided a generic critique regarding medical/satellite data (guess: 4).

---

### Decision · Program_Chairs · 2026-01-26

Accept (Poster)